# Probably Approximately Global Robustness Certification

Peter Blohm [1]   Patrick Indri [1]   Thomas Gärtner [1]   Sagar Malhotra [1]

## Abstract

We propose and investigate probabilistic guarantees for the adversarial robustness of classification algorithms. While traditional formal verification approaches for robustness are intractable and sampling-based approaches do not provide formal guarantees, our approach is able to efficiently certify a probabilistic relaxation of robustness. The key idea is to sample an $\epsilon$-net and invoke a local robustness oracle on the sample. Remarkably, the size of the sample needed to achieve probably approximately global robustness guarantees is independent of the input dimensionality, the number of classes, and the learning algorithm itself. Our approach can, therefore, be applied even to large neural networks that are beyond the scope of traditional formal verification. Experiments empirically confirm that it characterizes robustness better than state-of-the-art sampling-based approaches and scales better than formal methods.

## 1. Introduction

We present a novel sampling-based procedure to certify the *global robustness* of a classifier with high probability. Existing robustness verification methods are decision procedures that provide yes/no answers for a given robustness criterion, for a specific point in the input space. In general, however, robustness is a quantitative property that may depend on additional factors, such as the classifier's confidence in a given prediction. As a key contribution, we propose and investigate a novel notion of global robustness that quantifies the robustness of any point, given its *prediction confidence*. Our guarantee is obtained by checking local robustness on a *sufficiently large* sample of points. The size of the sample is quantified using a learning-theoretic construct, namely $\epsilon$-nets (Haussler & Welzl, 1986). Our approach is agnostic to the specific method used to perform local robustness

checks and, therefore, to the precise notion of robustness investigated. In this paper, we will consider both formal and adversarial methods to check local robustness.

Neural Network (NN) robustness is a major desideratum as, for non-robust networks, predictions can be drastically changed with only small perturbations to the input (Szegedy et al., 2014). This behavior can be detrimental in safety-critical applications like autonomous driving (Rao & Frtunikj, 2018) or image recognition tasks (Athalye et al., 2018). Hence, methods for evaluating whether NNs are robust to such perturbations are critical to enable their safe and reliable deployment. Significant research efforts have been invested to assess NN robustness, following two main lines of research:

- *Formal verification* methods have been used to provably certify local robustness of NNs (Katz et al., 2019; Wu et al., 2024; Xu et al., 2021). Recent results have extended these techniques to global robustness certification (Athavale et al., 2024) albeit only for NNs with up to a few hundred parameters.

- *Adversarial* methods rely on local optimization to assess whether NNs can withstand adversarial attacks (Goodfellow et al., 2015; Madry et al., 2018; Carlini & Wagner, 2017a). These methods easily scale to large networks, but generally do not provide formal guarantees on the global behavior of the network.

Our approach provides a quantitative characterization of robustness for the whole input space parameterized by the prediction confidence of each point. That is, for each point in the input space we are able to give a high-probability lower bound for its robustness. This only requires independently and identically distributed (iid) samples from the data distribution and access to a local robustness oracle. Such an oracle can be efficiently modeled using existing (formal or adversarial) methods for assessing local robustness. The sample size required by our approach is independent of the input dimensionality, the number of classes, and the learning algorithm itself. Our approach can provide distinct robustness guarantees *for each* confidence value after a *single* sampling procedure. This characterization can be used to infer high-probability lower bounds on the robustness of new data points, that were not part of the sampling procedure.

---

[1]TU Wien, Austria. Correspondence to: Peter Blohm <peter.blohm@tuwien.ac.at>.

*Proceedings of the 42nd International Conference on Machine Learning*, Vancouver, Canada. PMLR 267, 2025. Copyright 2025 by the author(s).

This paper is organized as follows. In Section 2, we present the relevant related work. In Section 3, we introduce the preliminaries. Our main contributions are described in Section 4 and Section 5. We present and discuss our experimental evaluation in Section 6. Finally, Section 7 contains concluding remarks.

## 2. Related Work

After the seminal work by Szegedy et al. (2014) showed that NNs are sensitive to adversarial examples, a number of techniques to find such examples were introduced. Many such approaches rely on gradient computation such as the Fast Gradient Signed Method (FGSM, Goodfellow et al., 2015) and an iterative adaptation of it called Projected Gradient Descent (PGD, Madry et al., 2018). Additionally, the C&W attack (Carlini & Wagner, 2017a) explicitly takes the distance to the original data point into account, to find a particularly close adversarial example. More recently, empirical approaches to assess (Webb et al., 2019; Baluta et al., 2021; Kim et al., 2023) as well as improve (Li et al., 2023; Kim et al., 2023) the robustness of NNs have been introduced, often building from the concept of adversarial training introduced by Goodfellow et al. (2015). While these approaches are applicable to large NNs and can be used to empirically assess robustness, they do not provide theoretical robustness guarantees for new points.

In an effort to obtain provable guarantees, a line of research has developed formal methods for the verification of NNs (Katz et al., 2019; Chen et al., 2021; Xu et al., 2020). While a formally verified NN is provably robust to input perturbations (Casadio et al., 2022; Meng et al., 2022), formal verification is limited to small NNs. In fact, it is generally hard to provide guarantees about the behavior of large networks (Katz et al., 2017) and it is, in particular, hard to detect *all* adversarial examples (Carlini & Wagner, 2017b). Other approaches have sought to provide *approximate* (Wu et al., 2020) or *statistical* (Webb et al., 2019; Cohen et al., 2019; Lecuyer et al., 2019) robustness guarantees, as well as to train NNs which are certifiably robust (Li et al., 2023; Sinha et al., 2018; Wang et al., 2018).

Recent work has specifically addressed global robustness for NNs. Athavale et al. (2024) and Indri et al. (2024) have recently extended existing formal verification methods to certify global robustness for *confident* predictions, where confidence is quantified using the softmax function. Similarly, Kabaha & Cohen (2024) focus on confident predictions using a margin-based notion of confidence. However, these methods are limited to NNs with only hundreds of parameters. In this paper, we introduce a probabilistic relaxation of the notion of global robustness of Athavale et al. (2024). We use $\epsilon$-nets (Haussler & Welzl, 1986) to provide high-probability guarantees for this notion of robustness.

## 3. Preliminaries

In this section, we will introduce basic notation and the necessary background for our theory. We will further introduce and motivate the notion of a *robustness oracle* and *prediction confidence*. These abstract concepts will allow our results to be easily adapted to different methods of assessing robustness, as shown in our experiments in Section 6.

### 3.1. Basic Notation

Consider a classification task with $n$ classes on a metric input space $\mathcal{X}$. We define a classifier as $f : \mathcal{X} \to \mathbb{R}^n$, where for any instance $\mathbf{x} \in \mathcal{X}$ the classifier returns a vector $f(\mathbf{x}) \in \mathbb{R}^n$. In this vector, the $i$-th component $f(\mathbf{x})_i$ represents the classifier's output for class $i$, like the output layer in a NN. We say that the predicted class for $\mathbf{x}$ is $\mathbf{class}(\mathbf{x}) = \arg\max_{i \in [1,n]} f(\mathbf{x})_i$. Furthermore, we say $\mathbf{conf}_f(\mathbf{x}) \in \mathbb{R}$ is the *prediction confidence* of $f$ for $\mathbf{class}(\mathbf{x})$. In the rest of this paper, we will assume that for a class $\mathbf{class}(\mathbf{x}) = c$ the confidence is given by the softmax function, i.e.,

$$\mathbf{conf}_f(\mathbf{x}) = \frac{\exp(f(\mathbf{x})_c)}{\sum_{j \in [1,n]} \exp(f(\mathbf{x})_j)} \qquad (1)$$

without restricting the generality of our statements. We assume that the data for the classification task follows a distribution $\mathcal{D}$ over $\mathcal{X}$. We write $X \sim \mathcal{D}$ to indicate that $X$ is a random variable drawn from $\mathcal{D}$, and $\mathbf{x} \sim \mathcal{D}$ to denote an observed realization of $X$. If clear from context, probabilities $\Pr_{\mathcal{D}}$ that are computed with respect to a distribution $\mathcal{D}$ will be written as $\Pr$.

### 3.2. Robustness

We say a classifier $f$ is *robust* around a point $\mathbf{x}$, if $\mathbf{class}(f(\mathbf{x}))$ remains constant within some *neighborhood* $\mathcal{N}(\mathbf{x}) \subset \mathcal{X}$ of $\mathbf{x}$. In many applications, $\mathcal{N}(\mathbf{x})$ is chosen to be an $L_\infty$-ball (Goodfellow et al., 2015; Madry et al., 2018) or another norm-bounded ball (Baluta et al., 2021; Carlini & Wagner, 2017a; Leino et al., 2021; Athavale et al., 2024).

How robust $f$ is around $\mathbf{x}$ is assessed by a *robustness oracle* $\mathbf{rob}_f(\mathbf{x})$. The robustness oracle $\mathbf{rob}_f(\mathbf{x})$ returns the distance $\rho$ to the closest counterexample $\mathbf{x}' \in \mathcal{N}(\mathbf{x})$ s.t. $\mathbf{class}(f(\mathbf{x}')) \neq \mathbf{class}(f(\mathbf{x}))$. Examples of robustness oracles $\mathbf{rob}_f$ include verification tools (Katz et al., 2017; Meng et al., 2022; Xu et al., 2021; Athavale et al., 2024), which offer exact results at high computational costs. For such exhaustive search methods, we can state $\mathbf{rob}_f(\mathbf{x}) = \min_{\mathbf{x}' \in \mathcal{N}(\mathbf{x})} \{ \|\mathbf{x} - \mathbf{x}'\| : \mathbf{class}(f(\mathbf{x})) \neq \mathbf{class}(f(\mathbf{x}')) \}$. In contrast, heuristic or adversarial methods (Goodfellow et al., 2015; Madry et al., 2018; Carlini & Wagner, 2017a) can efficiently find very close points with different class membership, which can be used as an upper bound for $\rho$. We will further discuss practically useful robustness oracles based on well-established methods in Section 6.

The following definition captures this oracle-based notion of robustness around a given point.

**Definition 3.1** (Local $\rho$-robustness). A classifier $f : \mathcal{X} \to \mathbb{R}^n$ is locally $\rho$-*robust* around $\mathbf{x}$ according to the robustness oracle $\mathbf{rob}_f$ if

$$\mathbf{rob}_f(\mathbf{x}) \geq \rho. \tag{2}$$

Based on this definition, we need to define a notion of *global* robustness for a given classifier $f$, so we can guarantee the classifier will behave robustly regardless of the input. A natural definition of *global $\rho$-robustness* would require local $\rho$-robustness for every possible input. However, enforcing this for all inputs in dense spaces like $\mathbb{R}^m$ would constrain the classifier to make constant predictions. To address this, a more useful notion of global robustness focuses on specific regions of interest, such as areas where the classifier's prediction confidence $\mathbf{conf}_f(\mathbf{x})$ is high (Leino et al., 2021; Athavale et al., 2024). More confident predictions require larger changes in the output of $f$ to alter the predicted class, so we consequently expect points classified with high confidence to be more robust. The following definition formalizes global robustness, based on a confidence threshold $\kappa$.

**Definition 3.2** (Global $\rho$-$\kappa$-robustness). A classifier $f : \mathcal{X} \to \mathbb{R}^n$ is *globally $\rho$-$\kappa$-robust* if it is locally $\rho$-robust for all $\mathbf{x} \in \mathcal{X}$ where the prediction confidence $\mathbf{conf}_f(\mathbf{x})$ is larger than $\kappa$.

$$\forall \mathbf{x} \in \mathcal{X} : \mathbf{conf}_f(\mathbf{x}) \geq \kappa \implies \mathbf{rob}_f(\mathbf{x}) \geq \rho. \tag{3}$$

Definition 3.2 is equivalent to the following statement, which states the absence of counterexamples:

$$\nexists \mathbf{x} \in \mathcal{X} : \mathbf{rob}_f(\mathbf{x}) < \rho \wedge \mathbf{conf}_f(\mathbf{x}) \geq \kappa. \tag{4}$$

Equation (4) highlights that a classifier is certified to be robust if no counterexample exists, that is, if no point with both high confidence and insufficient robustness is found by the oracle. However, certifying robustness in this manner is intractable in general (Katz et al., 2017). In this paper, we focus instead on bounding the probability of encountering counterexamples.

### 3.3. Epsilon Nets

Our robustness guarantees build on concepts from computational geometry and learning theory, and our definitions are based on Mitzenmacher & Upfal (2017).

**Definition 3.3** (Range space). Let $\mathcal{Y}$ be a (possibly infinite) set and $\mathcal{R}$ a family of subsets of $\mathcal{Y}$ called *ranges*. A range space is a tuple $(\mathcal{Y}, \mathcal{R})$.

**Definition 3.4** (VC Dimension, Vapnik & Chervonenkis (2015)). Let $(\mathcal{Y}, \mathcal{R})$ be a range space. The Vapnik-Chervonenkis (VC) dimension of $(\mathcal{Y}, \mathcal{R})$ is the size of the largest finite set $S \subseteq \mathcal{Y}$ such that for every subset $T \subseteq S$, there exists a range $R \in \mathcal{R}$ satisfying $R \cap S = T$. If no such maximum exists, the VC dimension is infinite.

**Definition 3.5** ($\epsilon$-net, Haussler & Welzl (1986)). Let $(\mathcal{Y}, \mathcal{R})$ be a range space and $\mathcal{D}$ be a probability distribution on $\mathcal{Y}$. A (finite) set $N \subseteq \mathcal{Y}$ is an $\epsilon$-net for $\mathcal{Y}$ with respect to $\mathcal{D}$ if, for every set $R \in \mathcal{R}$ such that $\Pr(R) \geq \epsilon$, the set R contains at least one point from $N$, i.e.,

$$\forall R \in \mathcal{R} : \ \Pr(R) \geq \epsilon \implies R \cap N \neq \emptyset. \tag{5}$$

$\epsilon$-nets provide a general notion of coverage. Our approach for certifying PAG-robustness relies on the following theorem for constructing $\epsilon$-nets from iid samples.

**Theorem 3.6** ($\epsilon$-nets from iid samples (Mitzenmacher & Upfal, 2017)). *Let $(\mathcal{Y}, \mathcal{R})$ be a range space with VC dimension $d$ and let $\mathcal{D}_{\mathcal{Y}}$ be a probability distribution on $\mathcal{Y}$. For any $0 < \delta, \epsilon \leq \frac{1}{2}$, an iid sample from $\mathcal{D}_{\mathcal{Y}}$ of size $s$ is an $\epsilon$-net for $\mathcal{Y}$ with probability at least $1 - \delta$ if*

$$s = \mathcal{O}\left( \frac{d}{\epsilon} \ln \frac{d}{\epsilon} + \frac{1}{\epsilon} \ln \frac{1}{\delta} \right) \tag{6}$$

While the big-$\mathcal{O}$ notation of this result is convenient for theoretical asymptotic analyses, it neglects constant factors that may be significant in practice. In their proof, however, Mitzenmacher & Upfal (2017) show that Theorem 3.6 always holds if $s$ satisfies the inequality in the following restatement.

**Proposition 3.7** ($\epsilon$-nets from iid samples with constants). *Let $(\mathcal{Y}, \mathcal{R})$ be a range space with VC dimension $d$ and let $\mathcal{D}$ be a probability distribution over $\mathcal{Y}$. For any $0 < \epsilon, \delta < \frac{1}{2}$, an iid sample from $\mathcal{D}_{\mathcal{Y}}$ of size $s$ is an $\epsilon$-net for $\mathcal{Y}$ with probability at least $1 - \delta$ if*

$$s \geq \frac{2}{\ln(2)\epsilon} \left( \ln \frac{1}{\delta} + d\ln(2s) - \ln\left(1 - e^{-s\epsilon/8}\right) \right). \tag{7}$$

*Proof.* See Appendix A. □

We refer to the smallest integer $s$ that satisfies the inequality in Proposition 3.7 as $s(\epsilon, \delta, d)$, and obtain it by computing

$$\min_{s \in \mathbb{N}} \left\{ s \geq \frac{2}{\ln(2)\epsilon} \left( \ln \frac{1}{\delta} + d\ln(2s) - \ln\left(1 - e^{-s\epsilon/8}\right) \right) \right\}. \tag{8}$$

To obtain sample complexities, we find $s(\epsilon, \delta, d)$ from Equation (8) using numerical methods.

## 4. PAG Robustness

In this section, we present our main theoretical results. We show how to get high-probability certificates for the following relaxation of global $\rho$-$\kappa$-robustness (Definition 3.2).

**Definition 4.1.** Given a data-distribution $\mathcal{D}$, a classifier $f : \mathcal{X} \to \mathbb{R}^n$ is *approximately globally robust* according to a robustness oracle $\mathbf{rob}_f$ if, for a given confidence $\kappa$, robustness $\rho$, and $X \sim \mathcal{D}$, we have that

$$\Pr\left(\mathbf{rob}_f(X) < \rho \mid \mathbf{conf}_f(X) \geq \kappa\right) < \epsilon. \quad (9)$$

where $0 < \epsilon < 1$ is a chosen parameter.

Our relaxation states that all $\kappa$-confident predictions of $f$ will also be at least $\rho$-robust, with high probability. This probabilistic statement is inspired by the work of Athavale et al. (2024), where the confidence threshold is introduced as a mechanism for $f$ to abstain from prediction. A key difference to their work is that we are able to decide the probabilistic statement for all values of $(\rho, \kappa)$ simultaneously. For this, we devise on a sampling-based methodology using $\epsilon$-nets to obtain the bound in Equation (9).

We rephrase the left-hand side of Equation (9) as

$$\frac{\Pr\left(\mathbf{rob}_f(X) < \rho \wedge \mathbf{conf}_f(X) \geq \kappa\right)}{\Pr\left(\mathbf{conf}_f(X) \geq \kappa\right)} \quad (10)$$

to then separately provide a lower bound for the numerator (using Proposition 4.2) and an upper bound for the denominator (using Lemma 4.3). Both of these bounds can then be realized with the same iid sample. In the rest of this section, we introduce a set of novel definitions required for our results and then proceed with the bounds themselves.

### 4.1. Quality Space

An important aspect of our method is that our guarantees are independent of the properties of the input space and of the classifier $f$. This is because our guarantees only consider the two properties of confidence and robustness of a given point in the input space. In this section, we formalize this idea. We define a function $q$ that maps a given point $\mathbf{x} \in \mathcal{X}$ to its robustness-confidence tuple:

$$q(\mathbf{x}) \mapsto (\mathbf{rob}_f(\mathbf{x}), \mathbf{conf}_f(\mathbf{x})). \quad (11)$$

As $q(\mathbf{x})$ captures precisely the characteristics, or *qualities*, of each point $\mathbf{x}$ that are relevant for our guarantees, it is useful to introduce the concept of a *quality space* $\mathcal{Q}$. The quality space is a 2-dimensional real space defined by the outputs of the map $q$. For a given pair $(\rho, \kappa)$, we say $\mathbf{x}$ is a *counterexample* to global $\rho$-$\kappa$-robustness if $q(\mathbf{x}) \in R(\rho, \kappa)$, where

$$R(\rho, \kappa) := \{(\rho', \kappa') \in \mathbb{R}^2 : \rho' < \rho \wedge \kappa' \geq \kappa\}. \quad (12)$$

The quality space allows us to easily define and detect whether $f$ is robust or not. Each $R(\rho, \kappa)$ corresponds to

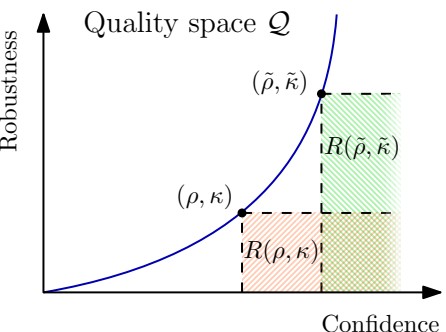

*Figure 1.* Visualization of the quality space $\mathcal{Q}$ and of the region of counterexamples defined in Equation (12) for two possible pairs of robustness-confidence values $(\rho, \kappa)$ and $(\tilde{\rho}, \tilde{\kappa})$.

the intersection of two-axis aligned half-spaces in $\mathcal{Q}$, as illustrated in Figure 1.

We define the family of these counterexample ranges over all possible $(\rho, \kappa)$ as

$$\mathcal{R} = \{R(\rho, \kappa) : (\rho, \kappa) \in \mathbb{R}^2\}. \quad (13)$$

Under a data distribution $\mathcal{D}$, we have

$$\begin{aligned}
\Pr(R(\rho, \kappa)) &= \Pr\left(q(X) \in R(\rho, \kappa)\right) \\
&= \Pr\left(\mathbf{rob}_f(X) < \rho \wedge \mathbf{conf}_f(X) \geq \kappa\right).
\end{aligned} \quad (14)$$

Given a set $N \subseteq \mathcal{X}$, we use $q(N)$ to denote the set $q(N) = \{q(\mathbf{x}) \in \mathcal{Q} : \mathbf{x} \in N\}$. With these concepts, we can obtain sample complexities independent from the dimensionality of $\mathcal{X}$. We obtain an iid sample $N$ in $\mathcal{X}$, map it to $\mathcal{Q}$, and interpret $q(N)$ as $\epsilon$-net over the range space $(\mathcal{Q}, \mathcal{R})$. This range space is equivalent to the intersection of two axis-aligned half-spaces in $\mathbb{R}^2$, with a VC-dimension $d = 2$.

### 4.2. Bounding the Joint Probability

In this section, we define a method that can check under which conditions and for which $(\rho, \kappa)$ the inequality

$$\Pr\left(\mathbf{rob}_f(X) < \rho \wedge \mathbf{conf}_f(X) \geq \kappa\right) < \epsilon \quad (15)$$

holds true. For our guarantee, we consider an iid sample $N$ from the data distribution $\mathcal{D}$. If our iid sample is an $\epsilon$-net, it intersects *all* $\epsilon$-likely ranges. Therefore, if no counterexample to $\rho$-$\kappa$-robustness is found in an $\epsilon$-net over $(\mathcal{Q}, \mathcal{R})$, then $R(\rho, \kappa)$ is known to be less than $\epsilon$-likely. The following proposition formalizes this idea.

**Proposition 4.2.** *Let $\mathcal{D}$ be a data distribution and $f : \mathcal{X} \to \mathbb{R}^n$ be a classifier. Given an $\epsilon$-net $q(N)$ over the range space $(\mathcal{Q}, \mathcal{R})$, then for a given confidence $\kappa$ and robustness radius $\rho$, we have that*

$$q(N) \cap R(\rho, \kappa) = \emptyset \implies \Pr\left(R(\rho, \kappa)\right) < \epsilon \quad (16)$$

where $R(\rho, \kappa)$ denotes the set of counterexamples to $\rho$-$\kappa$-robustness.

*Proof.* As $q(N)$ is an $\epsilon$-net, it holds that

$$\forall R \in \mathcal{R} : \Pr(R) \geq \epsilon \implies q(N) \cap R \neq \emptyset. \quad (17)$$

By contraposition, this statement is equivalent to

$$\forall R \in \mathcal{R} : \Pr(R) < \epsilon \impliedby q(N) \cap R = \emptyset. \quad (18)$$

Then, $q(N) \cap R(\rho, \kappa) = \emptyset \implies \Pr(R(\rho, \kappa)) < \epsilon$. $\quad\square$

This result allows us to bound the numerator of Equation (9). Note that Proposition 4.2 allows us to give guarantees on the behavior of $f$. However, a bound on the joint probability (alone) may lead to vacuous guarantees. In fact, the probability in Equation (15) is trivially zero if we consider a confidence value $\kappa$ large enough such that $\Pr(\mathbf{conf}_f(X) \geq \kappa) = 0$. For this reason, we require additional information about $\Pr(\mathbf{conf}_f(X) \geq \kappa)$.

## 4.3. Bounding the prediction confidence

In this section, we provide a method to lower bound the probability of obtaining a given prediction confidence as

$$\Pr(\mathbf{conf}_f(X) \geq \kappa) \geq p_{\min}. \quad (19)$$

Equation (19) is implied by the following statement about the cumulative distribution function of confidence:

$$\Pr(\mathbf{conf}_f(X) \leq \kappa) \leq 1 - p_{\min} \quad (20)$$

The following auxiliary lemma provides a general method to obtain a bound for a given quantile of a random variable from an iid sample.

**Lemma 4.3.** *Let $K$ be a real-valued random variable, and $N$ be an iid sample of $K$ with $|N| = s$. Denote with $N_{(i)} \in \mathbb{R}$ the $i^{th}$-largest element in the sample. Then for parameters $1 > p \geq \frac{1}{2}$ and $0 < \delta < \frac{1}{2}$, with probability of at least $1 - \delta$, we have that*

$$\Pr(K \leq N_{(i)}) \leq p \quad (21)$$

*for any integer $i$ such that*

$$i < sp - \sqrt{2sp \ln\left(\frac{1}{\delta}\right)}. \quad (22)$$

*Proof sketch.* The number of elements in an iid sample that are smaller or equal to the $p$-quantile $K_p$ of $K$ is a random variable $I \sim \text{Binom}(s, p)$. We are interested in finding the largest integer $i$ such that $N_{(i)} \leq K_p$ holds with probability at least $1 - \delta$ and, therefore, the largest integer $i$ such that $\Pr(I \geq i) \geq 1 - \delta$. We use a Chernoff bound for the deviation of $I$ from its expected value $\mathbb{E}[I] = sp$ to obtain an integer $i$ that is a high-probability lower bound for $I$. $\quad\square$

We use Lemma 4.3 with $K = \mathbf{conf}_f(X)$ and $p = 1 - p_{\min}$ to obtain $\kappa$ values for which

$$\Pr(\mathbf{conf}_f(X) > \kappa) \geq p_{\min}. \quad (23)$$

As $\Pr(\mathbf{conf}_f(X) \geq \kappa) \geq \Pr(\mathbf{conf}_f(X) > \kappa)$, we arrive at Equation (19). For compactness, we use

$$i(s, p, \delta) := \max_{i \in \mathbb{N}} \left\{ i : i < sp - \sqrt{2sp \ln\left(\frac{1}{\delta}\right)} \right\} \quad (24)$$

to refer to the largest $i$ which respects the bound in Equation (22). In our sample, we refer to the confidence value corresponding to this $i$ as $\kappa_{\max}$. For confidence values $\kappa \leq \kappa_{\max}$, Lemma 4.3 allows us to state with high probability that $\Pr(\mathbf{conf}_f(X) > \kappa) \geq p_{\min}$. We cannot, however, make a statement about confidence values that were not observed frequently enough in the sample, i.e., for $\kappa > \kappa_{\max}$. This upper bound on confidence values being certified is natural, as our statement relies only on the observed sample.

## 4.4. PAG Robustness

We have now introduced all necessary tools and conditions to certify a classifier $f$ to be approximately globally robust. So far, we assumed an $\epsilon$-net in the quality space $\mathcal{Q}$ as given. We now discuss how to obtain such an $\epsilon$-net using Theorem 3.6 to derive our main contribution: an iid sampling-based procedure that allows us to certify the approximate global robustness of a classifier $f$ with high probability.

**Theorem 4.4** (PAG robustness). *Let $\mathcal{D}$ be a data distribution, $f : \mathcal{X} \to \mathbb{R}^n$ be a classifier and $\mathbf{rob}_f$ be a local robustness oracle. With $q(\mathbf{x})$ defined as in Equation (11), with parameters $0 < \epsilon, p_{\min}, \delta < \frac{1}{2}$. Consider a sample $N \sim \mathcal{D}$ with $|N| \geq s(\epsilon, \delta/2, 2)$ as per Equation (8) and an integer $i = i(|N|, 1 - p_{\min}, \delta/2)$ as per Equation (24). Then, with a probability of at least $1 - \delta$ the following implication holds for all $\rho$ and for all $\kappa \leq N_{(i)}$:*

$$\left( q(N) \cap R(\rho, \kappa) = \emptyset \right) \implies$$
$$\Pr\left( \mathbf{rob}_f(X) < \rho \mid \mathbf{conf}_f(X) \geq \kappa \right) < \frac{\epsilon}{p_{\min}}. \quad (25)$$

*That is, under the absence of counterexamples in $N$, the classifier $f$ can be certified to be* Probably Approximately Globally *(PAG) robust.*

We present a proof of Theorem 4.4 that builds on the following lemma, which reformulates Proposition 4.2 as conditional probability.

**Lemma 4.5.** *Let $X \sim \mathcal{D}$ be a random data point in $\mathcal{X}$, $f : \mathcal{X} \to \mathbb{R}^n$ be a classifier, and parameters $0 < \epsilon, p_{\min} \leq \frac{1}{2}$ be a parameter. Given an $\epsilon$-net $q(N)$ over the range space*

$(\mathcal{Q}, \mathcal{R})$, *for all* $\rho, \kappa$ *such that* $\Pr(\mathbf{conf}_f(X) \geq \kappa) \geq p_{\min}$, *it holds that*

$$\Big( q(N) \cap R(\rho, \kappa) = \emptyset \Big) \implies$$
$$\Pr\Big( \mathbf{rob}_f(X) < \rho \mid \mathbf{conf}_f(X) \geq \kappa \Big) < \frac{\epsilon}{p_{\min}}. \quad (26)$$

*Proof.* Let us define, for brevity, the two random events $E_r = \{\mathbf{rob}_f(X) < \rho\}$ and $E_c = \{\mathbf{conf}_f(X) \geq \kappa\}$. By Proposition 4.2 we know

$$q(N) \cap R(\rho, \kappa) = \emptyset \implies \Pr(E_r \wedge E_c) < \epsilon. \quad (27)$$

Furthermore, by the definition of conditional probability

$$\Pr(E_r \mid E_c) = \frac{\Pr(E_r \wedge E_c)}{\Pr(E_c)}. \quad (28)$$

When $\Pr(E_c) = \Pr(\mathbf{conf}_f(X) \geq \kappa) \geq p_{\min}$, it holds that

$$\Pr(E_r \mid E_c) \leq \frac{\Pr(E_r \wedge E_c)}{p_{\min}} < \frac{\epsilon}{p_{\min}}, \quad (29)$$

which completes the proof. $\qquad\square$

*Proof of Theorem 4.4.* We assume that $|N| \geq s(\epsilon, \delta/2, 2)$. Theorem 3.6 then implies that $q(N)$ is an $\epsilon$-net over $(\mathcal{Q}, \mathcal{R})$ with a probability of at least $1 - \delta/2$. We further assert $i \leq i(|N|, 1 - p_{\min}, \delta/2)$. Lemma 4.3 then implies that $\Pr(\mathbf{conf}_f(X) \geq N_{(i)}) \geq p_{\min}$ with a probability of at least $1 - \delta/2$ as well. We use the union bound to state that both conditions hold true with a probability of at least $1 - \delta$. Given that both $N_{(i)}$ is a valid quantile bound, i.e., $\Pr(\mathbf{conf}_f(X) \geq N_{(i)}) \geq p_{\min}$, and $\kappa \leq N_{(i)}$, then $\Pr(\mathbf{conf}_f(X) \geq \kappa) \geq p_{\min}$. The theorem then follows from Lemma 4.5. $\qquad\square$

In some settings, it might not be possible to directly sample from the data distribution, but rather from a distribution $\mathcal{D}'$ that is used to approximate the true distribution $\mathcal{D}$. Even in these settings we can transfer our guarantee, as long as we can quantify the difference between the two distributions.

**Theorem 4.6** (PAG under distribution shift). *Let $\mathcal{D}$ be a data distribution, $f : \mathcal{X} \to \mathbb{R}^n$ be a classifier and $\mathbf{rob}_f$ be some robustness oracle. Let $\mathcal{D}'$ be a distribution such that $\mathrm{TV}(\mathcal{D}, \mathcal{D}') = \Lambda$. Consider an iid sample $N \sim \mathcal{D}'^s$ and parameters $\epsilon$, $\delta$ and $p_{\min}$. Consider an $s$ that satisfies the conditions in Theorem 4.4 for $\mathcal{D}'$. Then, with probability at least $1 - \delta$*

$$\Big( q(N) \cap R(\rho, \kappa) = \emptyset \Big) \implies$$
$$\Pr_{\mathcal{D}}\Big( \mathbf{rob}_f(X) < \rho \mid \mathbf{conf}_f(X) \geq \kappa \Big) < \frac{\epsilon + \Lambda}{p_{\min} - \Lambda}. \quad (30)$$

*That is, in the absence of counterexamples $f$ can be certified to be PAG robust with respect to the data distribution $\mathcal{D}$.*

*Proof sketch.* The data processing inequality (Raginsky, 2014) can be used to show (Lemma A.2) that $\Lambda$ is an upper bound for the total variation distance of the distribution of instances sampled from $\mathcal{D}$ and $\mathcal{D}'$ when they are mapped into the quality space $\mathcal{Q}$. The numerator in the right-hand side can be derived by showing (Lemma A.3) that an $\epsilon$-net under $\mathcal{D}'$ is an $(\epsilon + \Lambda)$-net under $\mathcal{D}$. The denominator can be derived by observing (Proposition A.5) that the quantiles of $\mathcal{D}$ and $\mathcal{D}'$ can differ by at most $\Lambda$. The theorem follows (Appendix A) from these results. $\qquad\square$

In the following section, we describe how to use the theoretical results obtained so far and practically derive robustness lower-bounds for a given classifier.

## 5. Robustness Lower-Bounds

In the previous section, we described how to provide global robustness guarantees given an iid sample $N$ is found to be robust. Specifically, for a given choice of parameters $\epsilon$ and $\delta$, we bound the required size of $N$. This single sample $N$ can be used to evaluate the robustness with all tuples $(\rho, \kappa)$. A lower-bound on the robustness of a point $\mathbf{x}$ with prediction confidence $\kappa = \mathbf{conf}_f(\mathbf{x})$ can be obtained using the smallest observed robustness for a confidence of at least $\kappa$ in the sample $N$. We define a mapping function $M(\kappa) \mapsto \rho$ that maps a given confidence $\kappa$ to a corresponding robustness lower bound $\rho$:

$$M(\kappa) := \begin{cases} \begin{aligned} \min_{\mathbf{x} \in N} \quad & \mathbf{rob}_f(\mathbf{x}) \\ \text{s.t.} \quad & \mathbf{conf}_f(\mathbf{x}) \geq \kappa \end{aligned} & \text{if } \kappa \leq \kappa_{\max} \\ \text{UNDEFINED} & \text{else} \end{cases} \quad (31)$$

Equation (31) assures that no counterexample of $\rho$-$\kappa$-robustness is in $N$ for $\rho = M(\kappa)$, and that the returned $\rho$ satisfies all conditions of Theorem 4.4. The mapping function $M(\kappa)$ does not return a robustness lower-bound for confidence values larger than $\kappa_{\max}$. In this case, for the considered sample size the uncertainty for quantile estimation is too large to guarantee $\Pr(\mathbf{conf}_f(X) \geq \kappa_{\max}) > 0$, as described in Lemma 4.3. Note that, for $\kappa > \kappa_{\max}$, we can nevertheless guarantee that $\Pr\big( \mathbf{rob}_f(X) < \rho \wedge \mathbf{conf}_f(X) \geq \kappa \big) < \epsilon$. The map $M$ can be constructed from a sample $N$ in $\mathcal{O}(|N| \log(|N|))$ time, as detailed in Algorithm 1.

For a given mapping $M$, we now discuss how likely it is that our guarantees hold across all confidence values. This is not directly addressed by our analysis so far and requires to bound

$$\Pr\Big( \mathbf{rob}_f(X) < M\big( \mathbf{conf}_f(X) \big) \Big). \quad (32)$$

We can obtain a bound on this probability as a direct consequence of Proposition 4.2. In this sense, the $\rho$-$\kappa$-mapping

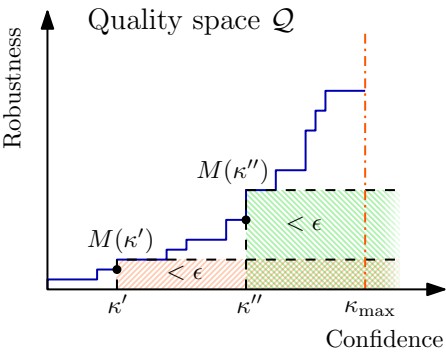

*Figure 2.* Visualization of the robustness lower-bound map $M(\kappa)$. Each rectangular region under the curve has a probability mass smaller than $\epsilon$. The yellow line represents the maximum confidence value above which $M(\kappa)$ is undefined.

$M$ is not only useful to obtain robustness lower-bounds for individual points, but can furthermore be used to bound the probability that for a point $\mathbf{x}$ sampled from $\mathcal{D}$, it holds that $\mathbf{rob}_f(\mathbf{x}) < M(\mathbf{conf}_f(\mathbf{x}))$.

**Proposition 5.1.** *Consider a classifier $f : \mathcal{X} \to \mathbb{R}^n$ and a $\rho$-$\kappa$-mapping $M$ constructed from an $\epsilon$-net $q(N)$ as in Equation (31). Let $|M|$ be the size of the codomain of $M$. Then*

$$\Pr\left(\mathbf{rob}_f(X) < M\left(\mathbf{conf}_f(X)\right)\right) < |M|\epsilon. \quad (33)$$

*Proof.* We use the properties of $\epsilon$-nets together with a union bound based on the size of the codomain of the mapping $M$. As $M$ consists, by construction, of $|M|$ discrete *steps*, $|M|$ ranges fully cover the area in the quality space below $M$. In the worst case, we have to assume their probability mass is additive and proceed with a union bound, proving our statement. □

In practical settings, $|M|$ is typically small; we empirically demonstrate this in Section 6. We illustrate the practical effectiveness of our method in experiments with NNs in the following section.

## 6. Experiments

In this section, we investigate the practical aspects of our theory. We aim to show that our results translate into practical settings and aim to answer the following research questions:

- **RQ1**: How can different methods for checking local robustness be modeled as oracles?

- **RQ2**: How well do our guarantees hold on unseen data in realistic, imperfect conditions?

- **RQ3**: How does the runtime of our verification procedure scale based on network size, parameter choices, and oracle?

- **RQ4**: How well can we capture qualitative differences in the behavior of different NNs?

We answer these questions by applying our procedure to the certification of NNs for MNIST (Deng, 2012) and CIFAR10 (Krizhevsky, 2009).

**Setup** We train four different network architectures (Ioffe & Szegedy, 2015; He et al., 2016; Mirman et al., 2018) on the two classification problems MNIST and CIFAR10, as illustrated in Table 1. For each architecture, we train, in a cross-validation setup, five instances of the network with standard training and five instances with TRADES (Zhang et al., 2019). We then use the respective validation split of the data to produce our guarantee: we imitate iid samples from the true data distribution by sampling with Gaussian noise from the validation data. As this approximation might not represent the true data distribution perfectly, the theoretical strictness of our guarantees might decrease, according to Theorem 4.6. Similar to this, in practical settings, true iid sampling is not always possible, and we investigate whether our guarantees appear to degrade noticeably.

*Table 1.* Overview of used NN architectures and oracles.

| Dataset | Architecture | Params | Oracle |
|---------|-------------|--------|--------|
| MNIST | FeedForward | 39 k | PGD, LiRPA |
|  | ConvBig | 1 663 k | LiRPA |
| CIFAR10 | ResNet20 | 272 k | PGD |
|  | VGG11_BN | 9 491 k | PGD |

We repeat the randomized verification procedure multiple times for each instance of the NNs with different robustness oracles. A detailed description of the training and verification procedure is provided in Appendix C.

**Robustness Oracles** We provide three robustness oracles, and conduct experiments with one based on adversarial techniques and one based on formal methods, as an answer to **RQ1**. Our adversarial oracle uses PGD (Madry et al., 2018), with a high number of small gradient steps, to report the closest counterexample found. For the experiments with PGD, we set $\epsilon = 10^{-4}$, $p_{\min} = 0.01$, $\delta = 0.01$ and thus sample $s(\epsilon, \delta/2, 2) = 989534$ images. We use PGD to *quantify* robustness, by performing many small gradient steps from each data point, and we measure the $L_\infty$ distance to the first adversarial example.

The two formal robustness oracles we provide are based on the NN verification tools Marabou (Katz et al., 2019) and

auto_LiRPA[1] based on LiRPA Xu et al. (2020) respectively. We will only conduct multiple experiments using auto_LiRPA, due to the high computational demand of Marabou in our setting. For both of these tools, we certify local robustness in $L_\infty$ hypercubes of fixed distances, and use binary search to find a lower bound of the radius where the model is robust. In our experiments with auto_LiRPA, we choose $\epsilon = 2.5 \cdot 10^{-3}$, $p_{\min} = 0.05$ and $\delta = 0.01$. We consequently sample $s(\epsilon, \delta/2, 2) = 31635$ images.

**Evaluation** After the sampling procedure is completed, we construct the map $M$ as in Equation (31) from the sampling split of our data. The map $M$ provides the robustness lower bound obtained with the sampling procedure. We are then interested in empirically checking whether the robustness lower bound we obtained holds on the unseen testing split, that is, whether the testing data respects our guarantees. To this end, we define the following estimators. For a given confidence value $\kappa$, we estimate the conditional probability $\Pr(\mathbf{rob}_f(\mathbf{x}') < M(\kappa) \mid \mathbf{conf}_f(\mathbf{x}') \geq \kappa)$ for $\mathbf{x}' \in D_{\text{test}}$, that is, the probability that the testing data *does not* respect the robustness lower-bound provided by $M$.

$$p_\kappa = \frac{|\{\mathbf{x}' : \mathbf{rob}_f(\mathbf{x}') < M(\kappa) \wedge \mathbf{conf}_f(\mathbf{x}') \geq \kappa\}|}{|\{\mathbf{x}' : \mathbf{conf}_f(\mathbf{x}') \geq \kappa\}|}, \tag{34}$$

where $\mathbf{x}' \in D_{\text{test}}$. To estimate the worst-case violation of our guarantees, we consider the $\hat{p} = \max_{\kappa \leq \kappa_{\max}} p_\kappa$ and check if $\hat{p} \leq \epsilon/p_{\min}$. We furthermore report

$$n_{\text{c}} = |\{\mathbf{x}' \in D_{\text{test}} : \mathbf{rob}_f(\mathbf{x}') < M(\mathbf{conf}_f(\mathbf{x}'))\}|. \tag{35}$$

This helps evaluate Proposition 5.1 empirically. As discussed in Proposition 5.1, $|M|$ denotes the number of steps in $M$, i.e., the size of its codomain. If Proposition 5.1 holds for our sample, we then expect $n_c \leq |D_{\text{test}}||M|\epsilon$. For both MNIST and CIFAR-10, $|D_{\text{test}}| = 10^4$ and thus $|D_{\text{test}}|\epsilon = 1$. We then expect $n_{\text{c}}/|M| < 1$ for PGD and $n_{\text{c}}/|M| < 10$ for auto_LiRPA.

**Results** Table 2 reports a summary of our experimental evaluation. We perform a total of 230 experiments on 40 NNs, checking our guarantees against the estimators described above. In the majority of trials (211/230) our estimators suggest that our guarantees hold across *all* confidence values. To answer **RQ2**, we discuss possible explanations for the 19 observed violations, besides the natural variance of our estimators defined in Equations (34) and (35).

In the few cases where our estimators indicate that our guarantees are violated on unseen data, the models under analysis still follow the overall trend described by our guarantees (see Figure 3), with only slight violations. Considering our results in more detail, we can observe that 5 of

the 6 runs with violations for the FeedForward network on MNIST originate in one single validation data split, as detailed in Appendix D. This could be, for instance, due to a distribution shift between this specific validation split under Gaussian noise and the true data distribution. A similar phenomenon can be observed for ResNet20 on CIFAR10.

With these possible explanations in mind, we observe a remaining total of $5/180 \approx 0.028$ trials, where our guarantees seem not to generalize to the unseen test data as expected. This empirical fraction is close to our choice of $\delta = 0.01$, and considering our simple approach of estimating the data distribution, our method can be considered relatively resilient to imperfect sampling settings.

To discuss **RQ3**, we inspect the runtimes in Table 2. We find that our approach is able to effectively scale to large models like the 10 million parameter VGG11_BN with strict parameter values for $\epsilon, \delta$ and $p_{\min}$, that provide strong guarantees. The runtime of our procedure consists mainly of the individual local robustness checks, so by relaxing $\epsilon, \delta$ and $p_{\min}$, the runtime can be easily adjusted to desired durations.

We plot the map $M$ and the testing split of CIFAR10 with VGG11_BN in Figure 3. To investigate our method on the more challenging examples, we choose to plot the results for one of the 3 experiments for which we observe $p_\kappa > 0.01$ for some $\kappa$. Even in this case, the lower bound provided by $M$ generally holds on test data and describes well the specific behavior of the networks, despite very few (7) exceptions. The bounds also allow for a clear differentiation between the formal robustness of adversarially and normally trained networks, as observable in Figure 4 in Appendix D too, which answers **RQ4** positively.

# 7. Conclusion

In this work, we use $\epsilon$-nets to devise a sampling-based approach to provide probabilistic guarantees on the global robustness of a given classifier. Our approach is agnostic to the specific robustness oracle and can thus be adapted to a variety of existing approaches. Our guarantees are conditioned on the confidence of a prediction to allow for a flexible characterization of robustness. In our experimental evaluation, we used PGD and auto_LiRPA to evaluate local robustness and showed how our method: (i) characterizes the behavior of networks trained with both standard and adversarial methods, and (ii) obtains useful global robustness guarantees that transfer effectively to unseen data.

In future work, we will use other informative properties in addition to confidence, such as the predicted class, to further refine our guarantees. Beyond this, more refined robustness oracles and sampling procedures can be used to further improve the scalability and practical resilience of the obtained guarantees. As our approach is agnostic to the

---

[1] github.com/Verified-Intelligence/auto_LiRPA

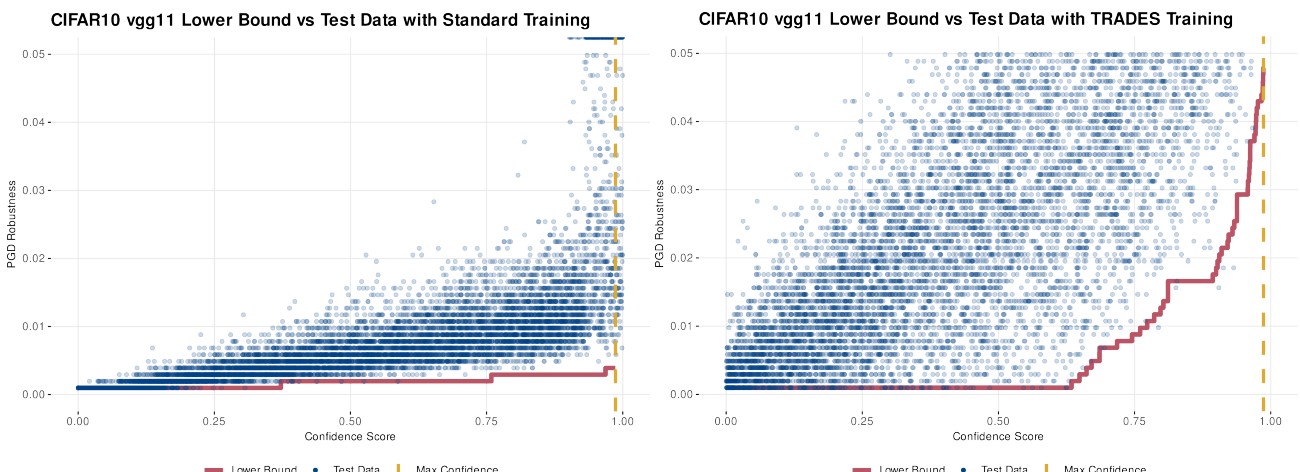

*Figure 3.* Scatter plot of the CIFAR-10 test dataset $D_{\text{test}}$, with $|D_{\text{test}}| = 10000$, in the quality space $\mathcal{Q}$ with VGG11_BN. The left network is trained with standard methods, the right network is trained robustly with TRADES. The red line depicts the lower bound obtained from validation sample $N$, with the parameters $\epsilon = 10^{-4}$, $\delta = p_{\min} = 0.01$, with $|N| = s(\epsilon, \delta/2, 2) = 989534$. On the right-hand side, 5 data points violate the lower bound. Note that, despite this apparent violation, $M$ tightly fits the test data and illustrates the contrasting robustness behaviors of the networks.

*Table 2.* Summary results for all experiments. We report *worst* results aggregated over three to five random seeds and over the different data splits used for the TRADES adversarial training. The subscripts indicate standard training (ST) or adversarial training (AT) with TRADES, and if the used oracle is PGD (P) or auto_LiRPA (L). For each experiment, we report the values of $\hat{p}$ and $n_c$, $|M|$, where bold numbers denote that the estimators are consistent with our guarantees *for all* the $\kappa \leq \kappa_{\max}$, *for all* the runs considered. Moreover, we report the number of individual "good runs" that are consistent with our guarantees even when considering the worst-case $\hat{p}$. Finally, we report the average runtime per verification run in minutes. More extensive results are available in Appendix D.

| Experiment | NN | $\hat{p} \cdot 10^3$ | $\epsilon/p_{\min} \cdot 10^3$ | $n_c$ | $|M|$ | good runs | runtime |
|---|---|---|---|---|---|---|---|
| MNIST$_{\text{ST,L}}$ | FeedForward | **0.00** | 50.00 | **0** | $13-15$ | 25/25 | 8.3 |
| | ConvBig | **0.00** | 50.00 | **0** | $5-5$ | 15/15 | 245 |
| MNIST$_{\text{AT,L}}$ | FeedForward | **1.30** | 50.00 | **7** | $6-9$ | 25/25 | 8.3 |
| | ConvBig | **3.39** | 50.00 | 44 | $3-4$ | 13/15 | 246 |
| MNIST$_{\text{ST,P}}$ | FeedForward | **0.18** | 10.00 | **2** | $38-46$ | 25/25 | 0.2 |
| MNIST$_{\text{AT,P}}$ | FeedForward | 16.50 | 10.00 | 4 | $14-22$ | 19/25 | 0.3 |
| CIFAR$_{\text{ST,P}}$ | ResNet20 | **1.35** | 10.00 | 8 | $3-3$ | 17/25 | 4.5 |
| | VGG11_BN | **0.00** | 10.00 | **0** | $4-5$ | 25/25 | 38.8 |
| CIFAR$_{\text{AT,P}}$ | ResNet20 | **1.51** | 10.00 | 4 | $24-42$ | 25/25 | 133.4 |
| | VGG11_BN | 17.00 | 10.00 | 9 | $26-60$ | 22/25 | 308 |

properties of the tested object, we will investigate other use cases where this approach to probabilistic verification could improve on the state of the art.

## Acknowledgements

This work was funded in part by the Vienna Science and Technology Fund (WWTF), project StruDL (ICT22-059); by the Austrian Science Fund (FWF), project NanOX-ML (6728); by the TU Wien DK SecInt and by the European Unions Horizon Europe Doctoral Network programme under the Marie-Skłodowska-Curie grant, project Training Alliance for Computational systems chemistry (101072930).

## Impact Statement

This paper presents work whose goal is to advance the field of Machine Learning. There are many potential societal consequences of our work, none of which we feel must be specifically highlighted here.

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

## A. Proofs

**Proposition 3.7** (ε-nets from iid samples with constants). *Let* $(\mathcal{Y}, \mathcal{R})$ *be a range space with VC dimension $d$ and let $\mathcal{D}$ be a probability distribution over $\mathcal{Y}$. For any $0 < \epsilon, \delta < \frac{1}{2}$, an iid sample from $\mathcal{D}_{\mathcal{Y}}$ of size $s$ is an $\epsilon$-net for $\mathcal{Y}$ with probability at least $1 - \delta$ if*

$$s \geq \frac{2}{\ln(2)\epsilon}\left(\ln\frac{1}{\delta} + d\ln(2s) - \ln\left(1 - e^{-s\epsilon/8}\right)\right). \tag{7}$$

*Proof.* We follow the "double sampling" argument from Mitzenmacher & Upfal (2017, Theorem 14.8). First, define $E_1$ as the random event that a sample $N$ of size $|N| = s$ is *not* an $\epsilon$-net:

$$E_1 = \left\{\exists R \in \mathcal{R} : \left(\Pr(X \in R) \geq \epsilon\right) \wedge \left(R \cap N = \emptyset\right)\right\}. \tag{36}$$

We aim to show $\Pr(E_1) \leq \delta$ for a large enough $s$. Consider then a second sample $T$ with $|T| = s$ and define $E_2$ as the random event that some range $R \in \mathcal{R}$ does *not* intersect $N$, but has a large intersection with $T$:

$$E_2 = \left\{\exists R \in \mathcal{R} : \left(\Pr(X \in R) \geq \epsilon\right) \wedge \left(R \cap N = \emptyset\right) \wedge \left(|R \cap T| \geq \frac{\epsilon s}{2}\right)\right\}. \tag{37}$$

As $\mathbb{E}(|R \cap T|) = \epsilon s$, then $\Pr\left(|R \cap T| \geq \frac{\epsilon s}{2}\right)$ should be large. Hence, $E_1$ and $E_2$ should have similar probability in total. Mitzenmacher & Upfal (2017) formalize this intuition with the following expression which considers a fixed range $R'$ such that $R' \cap N = \emptyset$ and $\Pr(X \in R') \geq \epsilon$. In particular, as $E_2 \subset E_1$ and consequently $E_2 = E_2 \cap E_1$, it follows that

$$\frac{\Pr(E_2)}{\Pr(E_1)} = \frac{\Pr(E_1 \cap E_2)}{\Pr(E_1)} = \Pr(E_2 \mid E_1) \geq \Pr\left(|T \cap R'| \geq \frac{\epsilon s}{2}\right). \tag{38}$$

For some fixed range $R'$, the random variable $S = |R' \cap T|$ follows a binomial distribution $\text{Bin}(s, p)$ with expectation $\mathbb{E}[S] = sp$. We can proceed with a Chernoff bound as

$$\Pr\left(|R' \cap T| \leq \frac{s\epsilon}{2}\right) \leq \Pr\left(|R' \cap T| \leq \frac{sp}{2}\right) \leq \exp\left(\frac{-sp}{8}\right) \leq \exp\left(\frac{-s\epsilon}{8}\right), \tag{39}$$

where we used the fact that $p \geq \epsilon$. While Mitzenmacher & Upfal (2017) relax this expression with $\exp(\frac{-s\epsilon}{8}) < \frac{1}{2}$. In the interest of tighter bounds, we continue instead our derivation without this relaxation. Thus,

$$\frac{\Pr(E_2)}{\Pr(E_1)} = \Pr(E_2 \mid E_1) \geq \Pr\left(|T \cap R'| \geq \frac{\epsilon s}{2}\right) \geq 1 - \exp\left(\frac{-s\epsilon}{8}\right) \implies \Pr(E_1) \leq \frac{\Pr(E_2)}{1 - \exp\left(\frac{-s\epsilon}{8}\right)}. \tag{40}$$

Next, we aim to bound the probability of $E_2$ using a larger event $E_2'$. Consider again some fixed range $R$ with

$$E_R = \{(R \cap N = \emptyset) \wedge (|R \cap T| \geq k)\} \tag{41}$$

with $k = \frac{\epsilon s}{2}$. We want to show that $\Pr(E_R)$ is small. To do so, consider a set of $2s$ elements and assume to partition it randomly into $N$ and $T$. $E_R$ captures the event that at least $k$ elements in $N \cup T$ intersect $R$ but none of these is in $N$. Of the $\binom{2s}{s}$ possible partitions of $N \cup T$, in exactly $\binom{2s-k}{s}$ of them, no element of $R$ is in $N$. Consequently,

$$\Pr(E_R) \leq \Pr(N \cap R = \emptyset \mid |R \cap (N \cup T)| \geq k) \tag{42}$$

$$\leq \frac{\binom{2s-k}{s}}{\binom{2s}{s}} = \frac{(2s-k)!s!}{(2s)!(s-k)!} = \frac{s(s-1)\cdots(s-k+1)}{(2s)(2s-1)\cdots(2s-k+1)} \tag{43}$$

$$\leq 2^{-k} = 2^{-\epsilon s/2}. \tag{44}$$

The last inequality introduces a further albeit small relaxation in the result, as $k \ll s$. We then finally consider the event $E_2'$ via the union bound over all the ranges $R \in \mathcal{R}$, that is,

$$E_2' = \{\exists R \in \mathcal{R} : (R \cap N = \emptyset) \wedge (|R \cap T| \geq \frac{s\epsilon}{2})\}. \tag{45}$$

We then use the Sauer-Shelah Lemma (Sauer, 1972; Shelah, 1972) to argue that we can consider at most $(2s)^d$ ranges when projecting $\mathcal{R}$ onto $N \cup T$. By the union bound we then have that $\Pr(E_2') \leq (2s)^d 2^{-s\epsilon/2}$. Finally, we arrive at

$$\Pr(E_1) \leq \frac{\Pr(E_2)}{1 - \exp\left(\frac{-s\epsilon}{8}\right)} \leq \frac{(2s)^d 2^{-s\epsilon/2}}{1 - \exp\left(\frac{-s\epsilon}{8}\right)} \leq \delta. \tag{46}$$

We can now simplify the last expression to obtain

$$(2s)^d 2^{-s\epsilon/2} \leq \delta \left(1 - \exp\left(\frac{-s\epsilon}{8}\right)\right) \tag{47}$$

$$d\ln(2s) + \left(\frac{-s\epsilon}{2}\right)\ln(2) \leq \ln(\delta) + \ln\left(1 - \exp\left(\frac{-s\epsilon}{8}\right)\right) \tag{48}$$

$$s \geq \frac{2}{\ln(2)\epsilon}\left(\ln\left(\frac{1}{\delta}\right) + d\ln(2s) - \ln\left(1 - \exp\left(\frac{-s\epsilon}{8}\right)\right)\right), \tag{49}$$

which concludes the derivation. $\qquad\square$

**Lemma 4.3.** *Let $K$ be a real-valued random variable, and $N$ be an iid sample of $K$ with $|N| = s$. Denote with $N_{(i)} \in \mathbb{R}$ the $i^{th}$-largest element in the sample. Then for parameters $1 > p \geq \frac{1}{2}$ and $0 < \delta < \frac{1}{2}$, with probability of at least $1 - \delta$, we have that*

$$\Pr(K \leq N_{(i)}) \leq p \tag{21}$$

*for any integer $i$ such that*

$$i < sp - \sqrt{2sp \ln\left(\frac{1}{\delta}\right)}. \tag{22}$$

*Proof.* Let $K_p \in \mathbb{R}$ be the $p$-quantile of $K$, i.e.,

$$\Pr(K \leq K_p) = p \tag{50}$$

The number of elements in an iid sample *smaller* than $K_p$ is a binomially distributed random variable $I \sim \text{Binom}(s, p)$. We aim to find an integer $i$ that results in a high-probability lower bound for $I$, that is, we look for the largest $i$ such that

$$\Pr(I > i) \geq 1 - \delta \tag{51}$$

or, equivalently,

$$\Pr(I \leq i) = \sum_{k=1}^{i}\binom{s}{k}p^k(1-p)^{s-k} < \delta. \tag{52}$$

There is no straightforward closed form expression to find this integer $i$. We proceed using Chernoff bounds for the deviation of $I$ from $\mathbb{E}[I] = sp$. Recall that Chernoff bounds for some relative deviation $\eta \in [0, 1]$ is defined as (Mitzenmacher & Upfal, 2017, Chapter 4)

$$\Pr(I \leq (1 - \eta)\mathbb{E}[I]) \leq \exp\left(\frac{-\eta^2\mathbb{E}[I]}{2}\right). \tag{53}$$

In our discrete setting, the relative deviation $\eta$ of $I$ from $\mathbb{E}[I]$ is defined as $\eta := \frac{sp-i}{sp}$. The Chernoff bound can then be written as

$$\Pr(I \leq i) \leq \exp\left(\frac{-(sp-i)^2}{2sp}\right) < \delta. \tag{54}$$

We now perform routine calculations to obtain an upper bound for $i$.

$$-\frac{(sp-i)^2}{2sp} < \ln(\delta) \tag{55}$$

$$\frac{(sp-i)^2}{2sp} > \ln\left(\frac{1}{\delta}\right) \tag{56}$$

$$sp - i > \pm\sqrt{2sp\ln\left(\frac{1}{\delta}\right)} \tag{57}$$

$$i < sp - \sqrt{2sp\ln\left(\frac{1}{\delta}\right)} \tag{58}$$

Given Equation (58) holds for a particular $i$, at least $i$ elements in a sample of size $s$ are smaller than $K_p$ with probability at least $1 - \delta$, i.e., $\Pr(N_{(i)} \leq K_p) \geq 1 - \delta$. Finally, the event $N_{(i)} \leq K_p$ implies $\Pr(K \leq N_{(i)}) \leq p$, which then holds true with a probability of at least $1 - \delta$ as well. □

**Definition A.1** (Total variation distance). Consider two distributions $\mathcal{D}, \mathcal{D}'$ over $\mathcal{X}$ and let $R \subseteq \mathcal{X}$ be a measurable set. The total variation (TV) distance between $\mathcal{D}$ and $\mathcal{D}'$ is defined as

$$\mathrm{TV}(\mathcal{D}, \mathcal{D}') = \sup_{R \subseteq \mathcal{X}} |\Pr_{\mathcal{D}}(R) - \Pr_{\mathcal{D}'}(R)|. \tag{59}$$

**Lemma A.2** (Total variation distance in the quality space $\mathcal{Q}$). *Consider two distributions $\mathcal{D}$ and $\mathcal{D}'$ over $\mathcal{X}$ and denote with $\mathrm{TV}(\mathcal{D}, \mathcal{D}')$ their total variation distance. For the function $q : \mathcal{X} \to \mathbb{R}^2$ that maps instances into the quality space $\mathcal{Q}$. We denote the distribution of the instances after they are mapped onto $\mathcal{Q}$ as $\mathcal{D}_q, \mathcal{D}'_q$. It then holds that*

$$\mathrm{TV}(\mathcal{D}_q, \mathcal{D}'_q) \leq \mathrm{TV}(\mathcal{D}, \mathcal{D}'). \tag{60}$$

*Proof.* For ease of presentation, we discuss a proof of the statement for more general conditions, which correspond to a version of the data processing inequality (Raginsky, 2014). Let $\mathcal{F}$ be a $\sigma$-algebra on the input space $\mathcal{X}$ and consider two probability measures $P$ and $Q$ on $(\mathcal{X}, \mathcal{F})$. Let $t : \mathcal{X} \to \mathcal{Y}$ be a measurable map and $\mathcal{G}$ be a $\sigma$-algebra on the space $\mathcal{Y}$. With slight abuse of notation we denote with $t(P)(B) \coloneqq P(t^{-1}(B)), \forall B \in \mathcal{G}$ the pushforward measure induced by $t$ on $(\mathcal{Y}, \mathcal{G})$. We do so analogously for $Q$ too. The total variation distance for probability measures is defined as

$$\mathrm{TV}(t(P), t(Q)) = \sup_{B \in \mathcal{G}} |t(P)(B) - t(Q)(B)| = \sup_{B \in \mathcal{G}} |P(t^{-1}(B)) - Q(t^{-1}(B))| \tag{61}$$

Note that for every measurable set $B \in \mathcal{G}$ in the measure space $(\mathcal{Y}, \mathcal{G})$, the preimage $t^{-1}(B) \in \mathcal{F}$ is a measurable set in $(\mathcal{X}, \mathcal{F})$. It therefore holds that

$$\sup_{B \in \mathcal{G}} |P(t^{-1}(B)) - Q(t^{-1}(B))| \leq \sup_{A \in \mathcal{F}} |P(A) - Q(A)| = \mathrm{TV}(P, Q), \tag{62}$$

from which it follows that

$$\mathrm{TV}(t(P), t(Q)) \leq \mathrm{TV}(P, Q). \tag{63}$$

The statement of lemma holds when $t$ is the function $q$ that maps to the quality space $\mathcal{Q}$, and $\mathcal{Y}$ is $\mathcal{Q}$. □

**Lemma A.3.** *Given two distributions $\mathcal{D}$ and $\mathcal{D}'$ on $\mathcal{X}$ such that $\mathrm{TV}(\mathcal{D}, \mathcal{D}') \leq \Lambda$, if $N$ is an $\epsilon$-net for $\mathcal{X}$ with respect to $\mathcal{D}'$, then it is an $(\epsilon + \Lambda)$-net for $\mathcal{X}$ with respect to $\mathcal{D}$.*

*Proof.* Using Definition A.1, $\mathrm{TV}(\mathcal{D}, \mathcal{D}') = \sup_{R \subseteq \mathcal{X}} |\Pr_{\mathcal{D}}(R) - \Pr_{\mathcal{D}'}(R)| \leq \Lambda$. Specifically, $\Pr_{\mathcal{D}}(R) \geq \epsilon + \Lambda$ implies that $\Pr_{\mathcal{D}'}(R) \geq \epsilon$. As $N$ is an $\epsilon$-net for $\mathcal{X}$ with respect to $\mathcal{D}'$, it intersects all $R \subseteq \mathcal{X}$ such that $\Pr_{\mathcal{D}'}(R) \geq \epsilon$, and it thus intersects all $R \subseteq \mathcal{X}$ such that $\Pr_{\mathcal{D}}(R) \geq \epsilon + \Lambda$. Therefore, $N$ is an $(\epsilon + \Lambda)$-net for $\mathcal{X}$ with respect to $\mathcal{D}$. □

**Lemma A.4.** *Let $K$ be a random variable sampled from $\mathcal{D}'$. Let $\mathcal{D}$ be another distribution such that $\mathrm{TV}(\mathcal{D}, \mathcal{D}') \leq \Lambda$, and $\Pr_{\mathcal{D}'}(K \leq \kappa) \leq p$ then we have that*

$$\Pr_{\mathcal{D}}(K \leq \kappa) \leq p + \Lambda \tag{64}$$

*Proof.* The proof follows from the fact that if $\mathrm{TV}(\mathcal{D}, \mathcal{D}') \leq \Lambda$, then probability of no event under $\mathcal{D}$ and $\mathcal{D}'$ can differ by more than $\Lambda$. $\qquad\square$

**Proposition A.5.** *Consider a random variable $K \sim \mathcal{D}'$ and let $N$, $N_{(i)}$, $p$, and $\delta$ be as in the setting for Lemma 4.3 so that $\mathrm{Pr}_{\mathcal{D}'}(K \leq N_{(i)}) \leq p$ holds with probability at least $1 - \delta$ for any integer $i$ such that $i < np - \sqrt{2np \ln{(1/\delta)}}$. Consider a distribution $\mathcal{D}$ such that $\mathrm{TV}(\mathcal{D}, \mathcal{D}') \leq \Lambda$. Then, under the distribution $\mathcal{D}$ and for the same $N$, $N_{(i)}$, $p$, and $\delta$, with probability of at least $1 - \delta$:*

$$\mathrm{Pr}_{\mathcal{D}}(K \leq N_{(i)}) \leq p + \Lambda. \tag{65}$$

*Proof.* Using Definition A.1, for any measurable set $R \subseteq \mathbb{R}^n$ it holds that $|\mathrm{Pr}_{\mathcal{D}}(R) - \mathrm{Pr}_{\mathcal{D}'}(R)| \leq \Lambda$. If we consider the event $\{K \leq N_{(i)}\} \subseteq \mathbb{R}^n$, with probability of at least $1 - \delta$ it holds that:

$$\mathrm{Pr}_{\mathcal{D}}(K \leq N_{(i)}) = \mathrm{Pr}_{\mathcal{D}'}(K \leq N_{(i)}) + \Lambda \overset{(\star)}{\leq} p + \Lambda, \tag{66}$$

where in $(\star)$ we used Lemma 4.3. $\qquad\square$

**Theorem 4.6** (PAG under distribution shift). *Let $\mathcal{D}$ be a data distribution, $f : \mathcal{X} \to \mathbb{R}^n$ be a classifier and $\mathbf{rob}_f$ be some robustness oracle. Let $\mathcal{D}'$ be a distribution such that $\mathrm{TV}(\mathcal{D}, \mathcal{D}') = \Lambda$. Consider an iid sample $N \sim \mathcal{D}'^s$ and parameters $\epsilon$, $\delta$ and $p_{\min}$. Consider an $s$ that satisfies the conditions in Theorem 4.4 for $\mathcal{D}'$. Then, with probability at least $1 - \delta$*

$$\Big( q(N) \cap R(\rho, \kappa) = \emptyset \Big) \implies$$
$$\mathrm{Pr}_{\mathcal{D}}\Big( \mathbf{rob}_f(X) < \rho \mid \mathbf{conf}_f(X) \geq \kappa \Big) < \frac{\epsilon + \Lambda}{p_{\min} - \Lambda}. \tag{30}$$

*That is, in the absence of counterexamples $f$ can be certified to be PAG robust with respect to the data distribution $\mathcal{D}$.*

*Proof.* We show that the two assumptions in Theorem 4.4 lead to the statement of the theorem if $\mathrm{TV}(\mathcal{D}, \mathcal{D}') = \Lambda$.

1. To satisfy assumption 1. in Theorem 4.4, $s \geq \frac{2}{\epsilon} \left( \log\left(\frac{4}{\delta}\right) + d \log(2s) \right)$, i.e., $N$ is an $\epsilon$-net for $\mathcal{X}$ with respect to $\mathcal{D}'$ with probability at least $1 - \delta/2$. Because of Lemma A.3, $N$ is also an $(\epsilon + \Lambda)$-net for $\mathcal{X}$ with respect to $\mathcal{D}$ with probability at least $1 - \delta/2$.

2. To satisfy assumption 2. in Theorem 4.4, $i < s(1 - p_{\min}) - \sqrt{2s(1 - p_{\min}) \ln\left(\frac{2}{\delta}\right)}$ for $i = |\{\mathbf{x} \in N : \mathbf{conf}_f(\mathbf{x}) < \kappa\}|$, i.e., $\mathrm{Pr}_{\mathcal{D}'}(\mathbf{conf}_f(X) \leq \kappa) \leq 1 - p_{\min}$ with probability at least $1 - \delta/2$. Using Proposition A.5 with $p = 1 - p_{\min}$, it follows that $\mathrm{Pr}_{\mathcal{D}}(\mathbf{conf}_f(X) \leq \kappa) \leq 1 - p_{\min} + \Lambda$ with probability at least $1 - \delta/2$.

By union bound, the probability that either condition does not hold on a given sample $N$ is at most $\delta/2 + \delta/2 = \delta$. Thus, with probability of at least $1 - \delta$, $N$ will be an $(\epsilon + \Lambda)$-net for $\mathcal{X}$ under $\mathcal{D}$ *and* the quantile bound holds under $\mathcal{D}$. The data processing inequality (Raginsky, 2014) can then be used to show (Lemma A.2) that $\Lambda$ is an upper bound for the total variation distance of the distribution of instances sampled from $\mathcal{D}$ and $\mathcal{D}'$ when they are mapped into the quality space $\mathcal{Q}$. With this, the theorem follows from the definition of $\epsilon$-nets and Lemma 4.5. $\qquad\square$

## B. Construction of Mapping

In this section, we expand on the definition of the mapping function $M$ in Equation (31). The construction of the map can be performed in a single pass over the sorted sample $N$, and is illustrated in Algorithm 1. This process is $\mathcal{O}(|N| \log(|N|))$, bound by the process of sorting. At inference time, $M(\kappa)$ returns the $\rho$ corresponding to the largest confidence $\kappa' \leq \kappa$.

The size of the mapping $|M|$, and the corresponding strictness of Proposition 5.1, can be controlled via quantization. Rounding down robustness radii reduces the size of the codomain of $M$, keeps the guarantees valid, and reduces the spatial requirements of the mapping.

## C. Details on Experimental Setup

In this section, we detail how the experiments were conducted. The full code to train, test and analyze the experiments is available at our repository.

---

**Algorithm 1** Obtain $\kappa$-$\rho$-mapping

---

**Input:** $\epsilon$-net $N$, confidence upper-bound $\kappa_{\max}$
**Output:** $\kappa$-$\rho$-mapping $M(\kappa)$
Let $M = \emptyset$
    Let $\kappa' = -\infty$
    `/* iterate through sample in order of increasing robustness ρ            */`
**for** $(\rho, \kappa) \in N$ in lexicographic order **do**
   **if** $\kappa' < \kappa \le \kappa_{\max}$ **then**
      $M = M \cup \{\kappa \mapsto \rho\}$ `// add new step from κ to ρ to the mapping`
      $\kappa' = \kappa$
   **end**
**end**
**return** $M$

---

## C.1. Software and Hardware Used

We use PyTorch to conduct our experiments and perform training with the MAIR (Kim et al., 2023) library for support of normal and adversarial training. In all instances, for TRADES, we chose the parameter $\beta = 6$. All the experiments were run on a single desktop machine equipped with an Intel i9-11900KF @ 3.50GHz CPU and a NVIDIA GeForce RTX 3080 GPU.

## C.2. Robustness Oracles

**PGD** We adapt the internal PGD function in MAIR to return the distance to adversarial examples. To get a fine-grained result, we perform many gradient steps with a small step size. For MNIST, we use a step size of $0.5/256$ and up to 200 steps to find an adversarial example. We project our examples to valid pixel values between 0 and 1. For CIFAR-10, we use a similar setup with a smaller step size of $0.1/256$ and up to 500 steps. We constrain adversarial images to have $L_\infty$ distance of at most 0.5 to the original data point.

**LIRPA** (Xu et al., 2020) offers a Python interface for NN robustness certification. Similar to Marabou, we perform some preprocessing on our MNIST models and use the bound propagation capabilities of LiRPA to obtain a bound. The LiRPA library lets us define a neighbourhood in $\mathcal{X}$ and gives bounds for the logits of $f$ in this neighbourhood. To check if the class of $f$ stays constant around a given point $\mathbf{x}$, with $\mathbf{class}(\mathbf{x}) = c$, we query LiRPA for an upper bound $\mathrm{UB}(\mathbf{x})$

$$\mathrm{UB}(\mathbf{x}) = \max_{\mathbf{x}' \in \mathcal{X}} \{ f(\mathbf{x}') - f_c(\mathbf{x}') : \|\mathbf{x} - \mathbf{x}'\|_\infty < \rho \}. \tag{67}$$

If this upper bound $\mathrm{UB}(\mathbf{x}) = 0$, then $f$ is robust around $\mathbf{x}$. It is worth noting that the bounds reported by LiRPA are not necessarily tight. This means that LiRPA *might underreport* robustness radii. Similarly to Marabou, we then use binary search to find the smallest value for $\rho$.

**Marabou** While omitted in the main paper due to the better scalability of `auto_LiRPA`, we present Marabou here as an additional example for a robustness oracle. We use the Python interface of the Marabou 2.0 NN verification tool to verify the following query for our network for a given data point $\mathbf{x} \in \mathcal{X}$ and a given robustness radius $\rho$:

$$\exists \, \mathbf{x}' \in \mathcal{X} : \|\mathbf{x} - \mathbf{x}'\|_\infty < \rho \wedge \mathbf{class}(\mathbf{x}) \neq \mathbf{class}(\mathbf{x}'). \tag{68}$$

If the formula is UNSAT, the network is locally $\rho$-robust. We perform a binary search to get a bounded estimate of the local robustness radius, and obtain a lower bound of the $L_\infty$ local robustness radius with 4 bits of precision, up to a value of 0.5. In Figure 6, we choose $\epsilon = 2.5/\ln(2) \cdot 10^{-3}$ and $\delta = 0.01$ and $p_{\min} = 0.05$, with a six-step binary search for $\rho$. This results in a sample requirement of $s(\epsilon, \delta/2, 2) = 21294$.

## C.3. NN Architectures

All network architectures were trained with the MAIR library (Kim et al., 2023), using a normal training procedure or TRADES (Zhang et al., 2019) with a $\beta = 6$. We trained 5 instances of each architecture on different splits of the training data. For the respective number of parameters, refer to Table 1.

**Feed Forward Network**   We considered three-layer, fully-connected ReLu networks with (768, 50, 10) neurons, trained on MNIST. Refer to our github repository and to Appendix D for the exact (hyper)parameter values.

**ResNet20**   We downloaded an untrained ResNet20 network from this Github repository. Refer to Appendix D for the exact hyperparameter values.

**ConvBig**   Similarly to the other architectures, we used an untrained ConvBig architecture (Mirman et al., 2018), but trained our instances independently.

**VGG11_BN**   This CIFAR10 network (Simonyan & Zisserman, 2015; Ioffe & Szegedy, 2015) is our largest architecture with approximately 10M trainable parameters.

### C.4. Datasets and Splits

We use MNIST and CIFAR10 for our experiments. We train five instances of each architecture in the manner of 5-fold cross-validation. We then use the respective 20% of the training data to sample data points with Gaussian noise added to the data points. The network does not see the validation split during training, and our guarantees are only obtained from the data in the test split. Finally, the test set, both unknown to the network and our verification procedure, is used to test the generalization of our guarantee.

To obtain the sampling datasets, we consider the following settings:

- For the PGD oracle, we choose the verification parameters $\epsilon = 10^{-4}$ and $\delta = p_{\min} = 0.01$. This results in a sample size of $s(\epsilon, \delta/2, 2) = 989534$.

- For the `auto_LiRPA` oracle, due to the more costly local verification procedure, we relax our parameters to $\epsilon = 2.5 \cdot 10^{-3}$, $\delta = 0.01$ and $p_{\min} = 0.05$. This results in a sample size of $s(\epsilon, \delta/2, 2) = 31635$.

For each of our experiments, the sampling dataset is then obtained by uniformly choosing data from the sampling split and adding Gaussian noise with a mean of 0 and a standard deviation of 8/256 for both.

## D. Additional Results

### D.1. Additional Results on MNIST

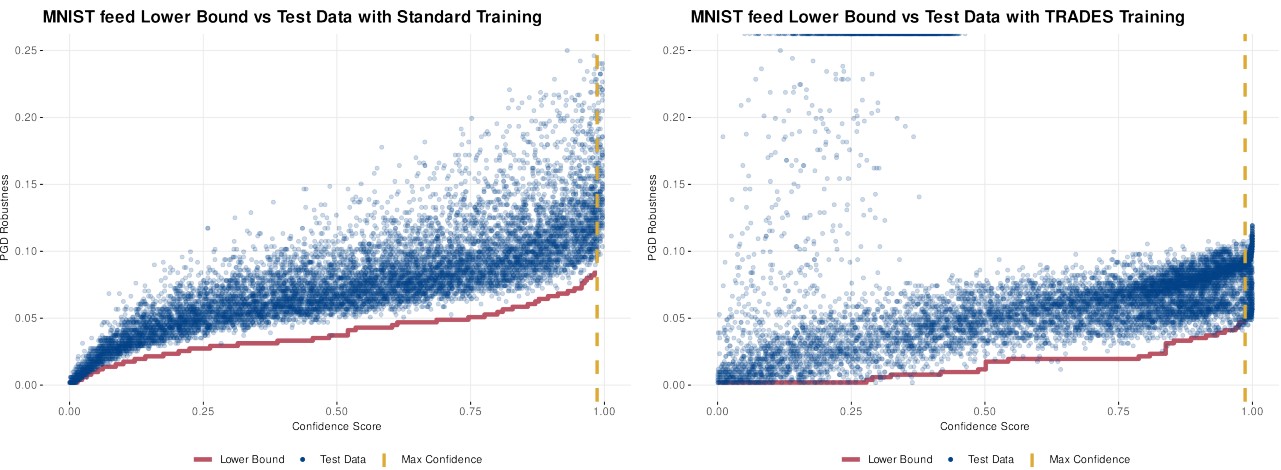

*Figure 4.* Scatter plot of the MNIST test dataset $D_{\text{test}}$, with $|D_{\text{test}}| = 10000$, in the quality space $\mathcal{Q}$ for our feed forward network. The left network is trained with standard methods, the right network is trained robustly with TRADES. Results for parameters $\epsilon = 10^{-4}$, $\delta = p_{\min} = 0.01$, with $|N| = s(\epsilon, \delta/2, 2) = 989534$.

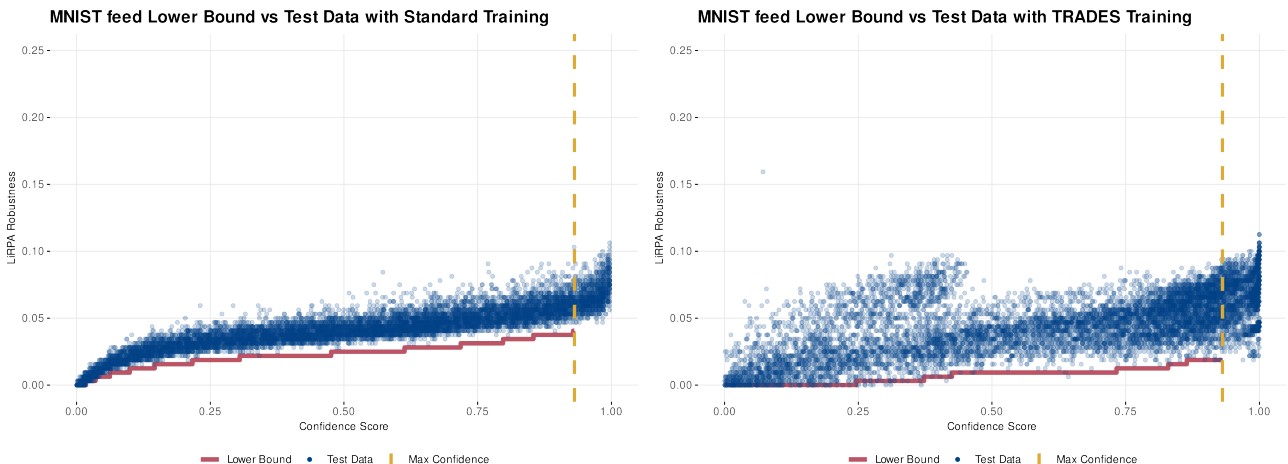

*Figure 5.* Scatter plot of the MNIST test dataset $D_{\text{test}}$, with $|D_{\text{test}}| = 10000$, in the quality space $\mathcal{Q}$ for our feed forward network. The left network is trained with standard methods, the right network is trained robustly with TRADES. Results for parameters $\epsilon = 2.5 \cdot 10^{-3}$, $\delta = p_{\min} = 0.01$, with $|N| = s(\epsilon, \delta/2, 2) = 31635$.

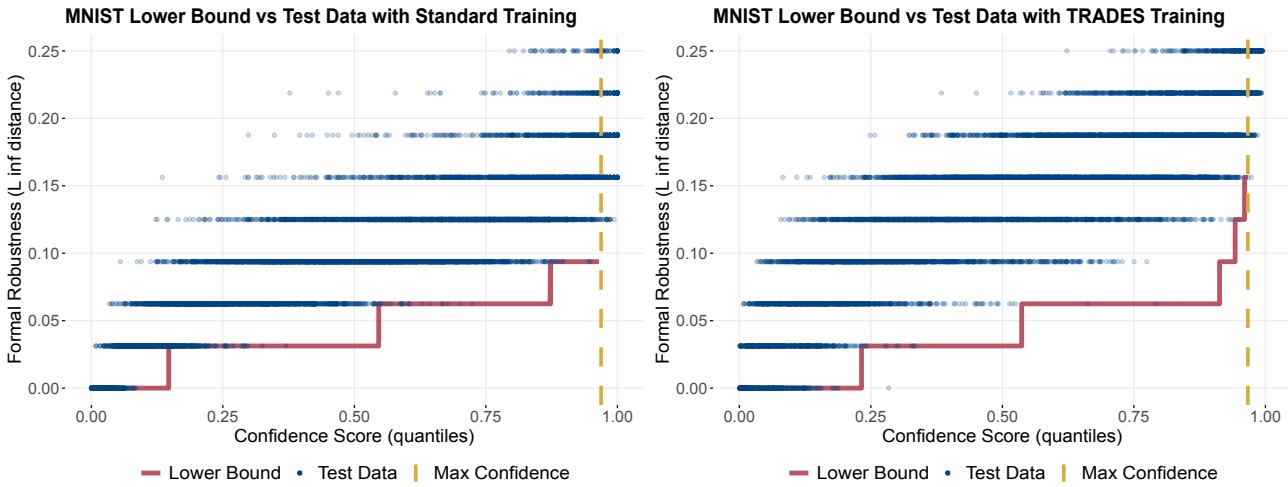

*Figure 6.* Scatter plot of the MNIST test dataset $D_{\text{test}}$, with $|D_{\text{test}}| = 10000$, in the quality space $\mathcal{Q}$ for our feed forward network. The left network is trained with standard methods, the right network is trained robustly with TRADES ($\beta = 2$). Results for parameters $\epsilon = 2.5/\ln(2) \cdot 10^{-3}$, $\delta = 0.01$ $p_{\min} = 0.05$, with $|N| = s(\epsilon, \delta/2, 2) = 21892$.

