# OpenReview forum: "Probably Approximately Global Robustness Certification"
_ICML.cc/2025/Conference — ICML 2025 poster_

### Official Review · Reviewer_8m6m · 2025-03-12

**Overall Recommendation:** 2

**Summary:**

The authors propose an algorithm that extends local robustness certification techniques to the entire input space with probabilistic guarantees. To achieve this, they introduce a novel approach for quantitatively characterizing a Deep Neural Network's (DNN's) robustness across the input space. Specifically, their method establishes high-probability lower bounds on robustness for each input point using only independent and iid samples and access to a local robustness oracle. The authors leverage existing local robustness testing oracles, such as adversarial attacks like PGD and formal local robustness verifiers, to determine the robust radius around a single sampled data point.

## Update after rebuttal
I thank the authors for their response. I have decided to maintain my scores and remain on the fence about this manuscript, as I see a mismatch between the theory and the proposed algorithm. The test dataset—or the samples obtained by perturbing it—may not truly represent i.i.d. samples from the global input distribution. Furthermore, relying on the test dataset inherits the same criticism faced by local robustness verification: it does not account for valid inputs (e.g., valid images of an airplane) that differ from those in the test set. Since the test dataset may not be a representative sample of all valid inputs for a given class, global robustness remains an important and unsolved problem.

**Claims And Evidence:**

**Strengths**
- The overall approach is mathematically well-motivated, and the use of $\epsilon$-nets to probabilistically characterize the DNN's robustness across the input space is an interesting theoretical idea.

**Weaknesses**
- The theoretical results and the proposed approach assume access to a sampling technique that can generate iid samples from the data distribution (e.g., MNIST/CIFAR10 images). However, the authors do not clarify how they ensure that their sampling method truly provides iid samples from the entire MNIST or CIFAR10 distribution. Instead, they rely on dataset samples (or dataset samples perturbed with Gaussian noise), meaning the approach inherits the same fundamental limitation as local robustness methods (**dependence on the Dataset**). Consequently, the guarantees remain dataset-dependent, which undermines the paper’s motivation of generalizing robustness guarantees to the entire input distribution.

- Building on the previous point, the proposed method appears to be a modification of **robust accuracy**, which measures the percentage of $L_{\infty}$ regions around test data points that are verified as robust in local robustness studies.


- In Section 3.1, the class of the sample $x$ is defined as $\text{class}(x) = \arg\max_{i \in [1, n]} f(x)_i$. Please correct me if I am wrong, but under this definition, even if the classifier $f$ consistently predicts the wrong class (one that does not match the ground truth) with high confidence, it would still be considered robust. Is this intentional? If so, does this mean that a classifier with very poor accuracy (making incorrect predictions that do not match the ground truth with high confidence) could still achieve a high robustness measure according to the proposed method?

**Essential References Not Discussed:**

I believe the authors have overlooked some of the most recent works on local robustness and local hyper-property verification (I referenced two from NeurIPS 2024 in the previous sections). There is also minimal discussion on GPU-accelerated Branch and Bound (BaB) based complete verifiers. I think the authors should consider using them for the CIFAR10 DNN experiments, which would help them apply formal methods to CIFAR10 networks as well.

**Experimental Designs Or Analyses:**

The experiments are limited to a single standard and adversarially trained MNIST and CIFAR10 network. Additionally, I have some concerns regarding the robustness oracles used—see `Methods and Evaluation Criteria`. Once the authors clarify these doubts, I may have follow-up questions regarding the experimental design and analysis.

**Methods And Evaluation Criteria:**

See the questions from the previous section and additionally:

- Exactly computing Eq. (30) is an NP-hard problem. Formal verifiers can compute a valid lower bound, while adversarial attacks provide an upper bound. However, both bounds can be imprecise. The authors should report both bounds to present a clearer picture.

- In lines 40–43, the authors claim: * That is, for each point in the input space we are able to give a high-probability lower bound for its robustness."* However, in the CIFAR10 experiments, they use only adversarial attacks, which compute an upper bound for Eq. (30)—a bound that can be significantly far from the optimal. Doesn't this contradict the claim made in lines 40–43? Please let me know if I am misunderstanding something.

**Other Comments Or Suggestions:**

I would suggest that the authors clarify the theoretical results in Sections 4 and 5, distinguishing between their original contributions and results derived from existing works or modifications of well-known results. This will make the paper much easier to read. For instance, Proposition 4.2 follows directly from Definition 3.5, and (may be wrong but) Lemma 4.3  appears to stem from existing results. Please explicitly mention these sources and highlight any specific extensions or modifications made to the existing results.

**Other Strengths And Weaknesses:**

Please refer to the `Claims And Evidence` and `Methods And Evaluation Criteria` section.

**Questions For Authors:**

Please refer to the `Claims and Evidence` and `Methods and Evaluation Criteria` sections. I am open to increasing my evaluation if my concerns regarding dataset dependency and the similarity to local robustness are adequately addressed.

**Relation To Broader Scientific Literature:**

I believe global robustness is a crucial problem for ensuring the safety of DNNs across the entire input distribution. However, my primary concern is that the guarantees obtained in this work, like those in local robustness studies, are essentially tied to the test dataset.

**Theoretical Claims:**

I have not carefully verified the detailed proofs in the Appendix. However, I have reviewed the proofs in the main paper and skimmed through those in the Appendix. They appear to be correct to me.

---

> ### Author Rebuttal · Authors · 2025-03-31
>
> Thank you very much for your thorough and insightful review.  We address your comments in the following.
>
> **Dependence on the dataset**.  We lift local robustness statements about a finite dataset to a (probabilistic) global robustness guarantee over the data distribution, i.e., to a robustness guarantee that generalizes with high probability.
> Hence, our approach directly addresses the dataset dependence of local robustness methods.
> For our theoretical statements, we indeed rely on the common assumption that we can get *unlabelled* iid samples from the data distribution.
> In our experiments, our sampling procedure may only approximate the data distribution.
> However, our evaluation shows that this approximation is sufficient to let our guarantee generalize to unseen data.
> This setup does not undermine our theoretical findings but rather demonstrates that our theory works in practice even in imperfect settings.
> This is also supported by Theorem 4.6.
>
> **Robust accuracy** is an empirical measure of model accuracy and local robustness.
> Given a set of points, robust accuracy quantifies, for these points *only*, how often a perturbation leads to misclassification.
> Our method is fundamentally different.
> Our approach does not aim to measure robustness directly but rather formalizes the conditions for generalization of robustness from a sample to the distribution.
> For this, we can provide guarantees for *any* given local robustness oracle, including one based on robust accuracy.
>
> **Robustness vs Accuracy.**  We consider robustness and accuracy to not necessarily be related.
> This is consistent with the literature on robustness verification (e.g. [a] from `gsc4`).
> If ground truth labels are available to the local robustness oracle, then our theory can accommodate a notion of robustness that considers accuracy as well.
>
> **Robustness lower-bound (Eq. 30)**. We use the expression “robustness lower-bound” to refer to the smallest distance to a counter-example reported by a chosen oracle $\mathbf{rob}_f$ on our sample $N$.
> For each point in the sample, the oracle reports a local robustness radius and in Eq. 30 we take the minimum of these reported values.
> Our global robustness guarantees are provided with respect to the attack scenario (e.g., a PGD attack) specified by the chosen robustness oracle, and Eq. 30 provides a valid lower-bound for this chosen scenario.
> While PGD may over-approximate the distance to the class boundary, if a PGD oracle is used then our method provides *valid guarantees* for adversarial robustness against PGD attacks (Fig. 4).
> We report lower bounds for both an adversarial and a formal oracle on the same data in Fig. 4 and 5.
> Compared to PGD, a formal verifier can provide more conservative bounds (Fig. 5).
>
> **Adversarial Attacks & Lower bounds (Line 40-43)** Our guarantees are provided with respect to a specific oracle and not necessarily with respect to the exact distance to the class boundary.
> Hence, our results for PGD are high probability robustness lower bounds for the distance required to reach an adversarial example using the PGD attack.
> We obtain these lower bounds by *exact computation of Eq. 30* for our sample $N$ with our chosen robustness oracle.
>
> **Experiments.** We have performed new experiments with larger architectures, including `convBig` [2], and with a new robustness oracle using the LiRPA library adopted by the CROWN verifiers.
> We provide details in the response to reviewer `md7J`.
> Both CIFAR and MNIST are commonly used benchmarks in the robustness literature [1, 2].
> The additional formal verification benchmarks in Zhou et al. [1] are out of the scope of our work as they do not consider a notion of robustness comparable to ours.
> In our experiments, we investigate networks of comparable size to [1,2] and often achieve better accuracy.
>
> **Other tools.** We run more experiments using recent libraries from the $\alpha$-$\beta$ CROWN tool (please see reply to `md7J`).
>
> **Theoretical results**. Thank you for the pointers, we will revise the section accordingly to highlight our contributions.
> Proposition 4.2 does follow from Definition 3.5.
> Lemma 4.3 is a small technical result stemming from rank statistics which does not directly follow from existing results to the best of our knowledge. However, we will be happy to cite any other relevant literature in your knowledge.
>
> We hope we have addressed your concerns and thank you again for your detailed review.
> We are very happy to engage in further discussion if you have more questions.

---

> > ### Comment · Reviewer_8m6m · 2025-04-04
> >
> > I will maintain my grades due to the following concerns and questions:
> >
> > - Assuming the samples generated by the authors are i.i.d. from the global data distribution, can the average robust accuracy (i.e., the sample mean) reported in local robustness studies serve as an approximation of the global average robust accuracy, provided the sample size is sufficiently large?
> >
> > - My main concern is that there is a mismatch between the theory and the actual experiments—specifically, that the test dataset, or the samples obtained by perturbing it, may not truly be i.i.d. samples from the global input distribution. Isn't this the key distinction between local and global robustness—where the goal is to assess robustness on entirely unseen data, which could be significantly different from the test dataset?

---

> > > ### Author Response · Authors · 2025-04-05
> > >
> > > We thank the reviewer for reading our rebuttal and engaging with us on the paper's methodology.
> > >
> > > In our experiments, we split the datasets into three parts: training, sampling and test.
> > > The *training* split is only seen during the NN training.  We sample from the sampling split with Gaussian noise to obtain our guarantees.
> > > The sampling split is the only split where Gaussian noise is used.
> > > **The test split is the official test dataset for MNIST and CIFAR10 respectively, without any perturbation or modification.**
> > > The test split is **unseen** by the training procedure and the procedure to obtain guarantees.
> > > The test split is used only to assess how our guarantees transfer on entirely unseen data.
> > > > “Isn't this the key distinction between local and global robustness—where the goal is to assess robustness on entirely unseen data, which could be significantly different from the test dataset?”
> > >
> > > Our experiments assess the robustness on test data that is **entirely unseen** and **adheres to the input data distribution**.
> > > The key idea of “global robustness guarantee” is that the guarantee generalizes to unseen/test data.
> > > Our method formalizes how many local assessments need to be performed to infer such a global robustness guarantee for a  **fixed input data distribution**.
> > >
> > > If test data were, instead, sampled from a **different distribution** with known total variation distance from the one used to obtain our guarantees, then Theorem 4.6 captures this scenario.
> > >
> > > If the test data is sampled from an ***arbitrarily different* distribution**, and we therefore have data which is arbitrarily different from the data we used to obtain our guarantees, then no statistical statement can be made, neither by our approach nor by any other learning theory  or statistics based methodology.
> > >
> > > Our notion of generalization from a sample to its fixed but unknown distribution is similar to the setting in PAC-learning [1,2] and also is similar to the intuition of your following question
> > >
> > > > “can the average robust accuracy (i.e., the sample mean) reported in local robustness studies serve as an approximation of the global average robust accuracy, provided the sample size is sufficiently large”,
> > >
> > > This is indeed possible. And our approach gives a bound on “sufficiently large sample size” for  our setting.
> > > However, the nature of the guarantee we introduce is completely different (and more useful) than a global average.
> > > First, robust accuracy is just one choice of a local robustness metric, **our guarantee works with any local notion of robustness.**
> > > Second, **average robust accuracy (or average of any local metric) can not be used to distinguish between the robustness of two points with different confidence values at test time.**
> > > We provide a high probability guarantee of a point’s robustness **given its prediction confidence**, which holds for the entire distribution and is much more nuanced than a global average.
> > >
> > > We thank the reviewer again for their analysis of our contribution. And we will be happy to further engage or clarify any  queries.
> > >
> > > [1]  L. G. Valiant: A Theory of the Learnable.1984
> > >
> > > [2] Mitzenmacher, M. and Upfal, E. Probability and Computing: Randomization and Probabilistic Techniques in Algorithms and Data Analysis, Second Edition, 2017 [Chapter 14, sec 14.4].

---

### Official Review · Reviewer_gTTU · 2025-03-13

**Overall Recommendation:** 2

**Summary:**

The authors propose a probabilistic framework to evaluate global robustness in neural networks. Global robustness for a neural network is defined such that a neural network is globally robust if it is robust at all confident predictions. The approach relies on \epsilon-nets and is evaluated on MNIST and CIFAR-10 datasets.

**Claims And Evidence:**

Some of the claims are not accurate. For instance:

- The authors claim that verification-based techniques for neural networks are limited to networks with small networks, where I guess small networks mean few hundreds of parameters. However, techniques such as CROWN have already scaled to larger networks than those considered in this paper [Wang, Shiqi, et al. "Beta-crown: Efficient bound propagation with per-neuron split constraints for neural network robustness verification." Advances in neural information processing systems 34 (2021): 29909-29921.]. Also, techniques like randomized smoothing [Cohen, Jeremy, Elan Rosenfeld, and Zico Kolter. "Certified adversarial robustness via randomized smoothing." international conference on machine learning. PMLR, 2019.], which could also provide guarantees (even if with some confidence), have already scaled to IMAGENET.

**Essential References Not Discussed:**

Most of the key references are mentioned, but, as I mentioned above for the Webb, S., et al. case or the literature in formal verification, some of them are not discussed properly.

**Experimental Designs Or Analyses:**

While experimental analysis seems sound, as mentioned above, I miss a comparison with comparable samples. Also, it would be important to see how the number of required samples changes with the confidence required.

**Methods And Evaluation Criteria:**

The evaluation performed by the authors makes sense. However, I miss an empirical comparison, for example, in terms of the number of required samples with comparable methods, such as [Webb, S., et al. "A statistical approach to assessing neural network robustness." Seventh International Conference on Learning Representations (ICLR 2019). International Conferences on Learning Representations, 2019], which seem to be applicable to the same specification.

**Other Comments Or Suggestions:**

N/A

**Other Strengths And Weaknesses:**

As mentioned above, it is difficult to evaluate the strengths of the paper without a proper comparison with the literature or more extensive experiments to see how the required number of samples changes.

**Questions For Authors:**

Apart from the ones mentioned above in terms of comparison with Webb, I have the following ones:

- How do you compute the VC dimension d for your problem with the various neural networks?

- Can your result also be applied if the data distribution support is unbounded?

- As some of your results are based on the fact that all samples are robust, how does your complexity in terms of the number of required samples change if the true robustness probability is not 100%, but say 99%?

**Relation To Broader Scientific Literature:**

Adversarial verification of neural networks is an important problem, and the probabilistic global notion of robustness considered here is surely important. However, as similar statistical approaches based on samples from the data distribution have already been used for similar problems, it would be important to have at least an empirical comparison with state-of-the-art to be able to judge the importance of the results.

**Theoretical Claims:**

Theoretical claims appear correct to me.

---

> ### Author Rebuttal · Authors · 2025-03-31
>
> Thank you very much for your review and comments.  We address them below.
>
> **Size of the certified NNs.** Techniques like CROWN and randomized smoothing do indeed scale up to large networks for *local robustness* certification.
> However, our work is concerned with using local robustness checks to provide *global robustness guarantees*.
> The per-example verification time reported by  Wang et al. is large for our application, as meaningful global guarantees require 10K-100K local robustness checks.
> Approaches that explicitly target global robustness (such as the ones referred to by reviewer `gsc4`, [a, b, c]) consider significantly smaller networks, with a number of parameters in the order of a few hundreds to a few thousands.
> Our approach scales significantly better than these approaches.
>
> **Randomized smoothing** does not address global robustness but provides certifications for a smoothed version of the original classifier around a given point. For certification, randomized smoothing can only provide local robustness guarantees, whereas we are interested in global robustness guarantees conditioned on the prediction confidence, that provably generalize to unseen data.
>
> **Comparison to Webb et al.** Our objective is fundamentally different to Webb et al.
> They provide a *statistical estimate* for the probability of failure  with respect to a given robustness property (as a boolean criterion, see eq. (2) in Webb et al.), with no guarantee on soundness of the estimate.
> In contrast, we provide a high probability *global guarantee* on robustness (as a metric) conditioned on the prediction confidence (eq. 24). Furthermore, the sample complexity of our approach does not depend on the dimensionality of the problem, and can be computed a priori (eq. 7).
> In contrast,  Webb et al. provide no formal sample complexity bounds. For any point they consider, they require a number of Metropolis-Hastings transitions that may increase with the dimensionality of the problem.
> Moreover, Webb et al. rely on a problem-dependent termination condition.
> We will add this discussion in the related work section.
>
> **Number of required samples.** The sample size required for our guarantees does not depend on the prediction confidence of the classifier and is instead fully captured by Equation 6 and 7.
> That is, the sample size depends only on the choice of parameters $\epsilon$ and $\delta$, and scales as $\mathcal O(\frac{1}{\epsilon}(\ln\frac{1}{\epsilon}+\ln\frac{1}{\delta}))$.
> This is one of the strengths of our approach, as the sample size does not depend on the dimensionality of the problem (see the remarks on the VC dimension below).
>
> **VC Dimension.** The VC dimension $d$ of the quality space is always equal to 2, for any given neural network, as it only depends on the property (i.e., robustness) we investigate.
> This is one of the main strengths of our approach, as the VC dimension of our problem does not depend on the choice of the learning algorithm, or on the specifics of the input data.
> The range space we consider is always constituted by the intersection of two axis-aligned half spaces in the 2-dimensional quality space, that is, it is constituted by the shaded regions in Figure 1.
>
> **Unbounded distribution support.** Yes, our method does not rely on the assumption that the data distribution is bounded.
> All our statements are non-parametric in nature. We do not rely on any specific notion of locality in the input space, besides the ones required by the local robustness oracle.
>
> **100% vs 99% robustness.** If we understand your question correctly, we want to clarify that the method does not depend on the fact that 100% of the observed samples are robust, and the sample complexity is not affected by this (see also above).
> Our method relies on the fact that if enough points ($\epsilon$-net many) are sampled and the minimum robustness value among them, for a given confidence $\kappa$, is found to be $\rho$, then we can say that with high probability a new point with confidence at least $\kappa$ will be at least $\rho$-robust.
> Our approach guarantees that such a statement indeed generalizes to test data.
>
>
> We hope our comments addressed the questions you raised. We are happy to engage in further discussion.

---

### Official Review · Reviewer_gsc4 · 2025-03-14

**Overall Recommendation:** 4

**Summary:**

This paper introduces a novel method for certifying the global robustness of neural networks (NNs). The method employs a sampling procedure to create an $\epsilon$-net, which is used along with a local robustness verifier to provide probabilistic guarantees on the robustness of the model, depending on its prediction confidence at a given point. Once the sampling procedure is performed, the approach can provide robustness guarantees for every point in the feature space without requiring the sampling procedure to be run again; in this sense, it provides global robustness guarantees. The approach assumes access to the true data distribution of instances in the feature space or a proxy of it (e.g., approximated with Gaussian noise). The experimental evaluation considers a feed-forward NN on MNIST and a ResNet-20 on CIFAR-10, demonstrating that the theoretical guarantees provided by the approach are also confirmed in practice. In particular, the probabilistic robustness guarantees hold with high probability in practice.

**Claims And Evidence:**

The claims of the paper are sufficiently supported by mathematical proofs and experimental evidence. The experimental evaluation convincingly shows that the theoretical bounds provided by the approach are reflected in practice. However, there are a few small points that the authors should discuss more in depth, in particular the approximation of the data distribution used by their approach and the influence of the sampling procedure on the total runtime of the method.


**Update after the authors' response**: the authors have satisfactorily answered to my questions and clarified the minor points.

**Essential References Not Discussed:**

There are no essential references omitted by the authors. However, they could make their discussion of related work more comprehensive by considering additional relevant approaches (see the _Relation To Broader Scientific Literature_ section).

**Update after the authors' response**: the authors have satisfactorily answered to my questions and discussed the suggested references.

**Experimental Designs Or Analyses:**

The experimental results presented in the paper support the claim regarding the validity of the guarantees on global robustness provided by the proposed approach. Figure 3 shows that the test data closely follows the mapping used by the approach to relate confidence values to lower bounds on robustness. Additionally, Table 1 demonstrates that, in the majority of runs, the global robustness guarantees provided by the approach hold even for unseen test data. However, there are some exceptions where this may not hold, and the authors have detailed the reasons behind these cases.

**Methods And Evaluation Criteria:**

The considered models (a feed-forward neural network and ResNet-20) and datasets (MNIST and CIFAR-10) are reasonable and in line or larger than the ones considered in previous related work (like [a]). Moreover, the chosen evaluation criterias are comprehensive to assess that the provided guarantees are satisfied and describe the behavior of the networks well.

[a] Athavale et. al., Verifying global two-safety properties in neural networks with confidence, CAV, 2024.

**Other Comments Or Suggestions:**

The authors should consider strengthening the discussion of related work by including works [a], [b] and [c] and discussing how the verification approach and/or the global robustness property considered in the current paper differ from those proposed in these three works.

[a] Kabaha and Cohen, Verification of Neural Networks’ Global Robustness, Proc. ACM Program. Lang., 2024.

[b] Wang et. al., Efficient Global Robustness Certification of Neural Networks via Interleaving Twin-Network Encoding, IJCAI, 2023.

[c] Calzavara et. al., Beyond Robustness: Resilience Verification of Tree-Based Classifiers, Computers&Security, 2022.


**Update after the authors' response**: the authors have satisfactorily discussed the suggested references.

**Other Strengths And Weaknesses:**

## Strengths

- This is the first approach that provides probabilistic approximate guarantees on the global robustness of neural networks, which can also scale to larger networks than those considered in previous work.
- The adopted notion of global robustness is intuitive and reasonable.
- Good theoretical foundation, including proofs of the statements that support the approach.
- The experimental evaluation is convincing.

## Weaknesses

- The approach works with both the original data distribution of the samples in the feature space and an approximation of it. However, the quality of the approximation used in the experimental evaluation is not clearly discussed.
- Missing details on how the sampling procedure influences the runtime of the approach.
- Related work discussion can be improved.

**Update after the authors' response**: the authors have satisfactorily addressed these weaknesses. Thus, I will raise my score.

**Questions For Authors:**

- In Section 6 (Experimental evaluation), the authors states that `we discuss how the data distribution $D$ can be approximated sufficiently well for our purposes with Gaussian noise.`. This point is important for the inner workings of the proposed certification approach, but I do not see any comments or results highlighting that the data distribution is well approximated. Could you please provide a comment on this and offer evidence? Have I missed something?
- The proposed approach requires sampling a large number of instances to obtain global robustness guarantees, e.g., 21,892 instances for the MNIST dataset. How does this step impact the total runtime of your approach? For which specific steps is the runtime reported in Tables 2, 3, and 4 computed?


**Update after the authors' response**: the authors have satisfactorily answered to these questions.

**Relation To Broader Scientific Literature:**

This work is closely related to [a], since both address the global robustness property while considering the confidence of the verified model, i.e., global robustness is verified for inputs where the model exhibits a certain level of confidence in its predictions. While the approach in [a] proposes a method that provides deterministic guarantees on global robustness, the method presented in this paper offers probabilistic guarantees. Although probabilistic guarantees are looser than deterministic ones, this choice enables the proposed method to scale to larger models and leverage standard verifiers for local robustness.

Other definitions of global robustness and verification methods have been proposed in [b], [c], and [d], since there is not a one-size-fits-all definition of global robustness in the literature at the moment. The authors should also discuss these approaches in relation to the ones they consider.

[a] Athavale et. al., Verifying global two-safety properties in neural networks with confidence, CAV, 2024.

[b] Kabaha and Cohen, Verification of Neural Networks’ Global Robustness, Proc. ACM Program. Lang., 2024.

[c] Wang et. al., Efficient Global Robustness Certification of Neural Networks via Interleaving Twin-Network Encoding, IJCAI, 2023.

[d] Calzavara et. al., Beyond Robustness: Resilience Verification of Tree-Based Classifiers, Computers&Security, 2022.

**Update after the authors' response**: the authors have satisfactorily answered to my questions and discussed the suggested references.

**Theoretical Claims:**

I checked the proof of Proposition 4.2, that serves as a building block for bounding the joint probability that the classifier provides a prediction above a certain threshold without being robust on the sample. I also checked the proofs of Lemma 4.3, which establishes a lower bound on the probability of obtaining a given (or bigger) prediction confidence by the classifier, and Lemma 4.5.

---

> ### Author Rebuttal · Authors · 2025-03-31
>
> Thank you for your thorough review and for the interesting and relevant references.
>
> **Quality of distribution approximation.** In the paper, "sufficiently well for our purpose" is intended to reflect that such an approximation is able to provide guarantees that generalize to unseen data. To avoid any misunderstanding, we will further elaborate on this in the paper.
> Getting enough representative samples from the true data distribution can be challenging in the real-world.
> However, our experiments demonstrate that the simple sampling procedure we use allows us to obtain global guarantees that consistently generalize to unseen data, despite possible deviations introduced by the sampling procedure itself.
> The fact that our (Gaussian-noise-based) sampling procedure is sufficient substantiates the practical effectiveness of our approach.
>  We also refer you to our answer to a similar question of reviewer `8m6m`.
>
> **Impact of sampling on runtime.** The influence of the sampling procedure itself on the runtime is negligible compared to the time spent performing the local robustness checks with the oracle.
> Overall, the runtime of the approach is linear in the sampling complexity in Equation 7 which, in turn, only depends on the chosen parameters $\epsilon$ and $\delta$, and not on the characteristics of the learning algorithm used or on the input dimensionality.
>
> **Runtimes in Tables.** Thank you for pointing this out, we will make it more explicit in the tables.
> The runtime reported is the runtime to produce our guarantees from the set of already sampled points, and thus includes evaluating their class, confidence and robustness according to the local robustness oracle.
> The sampling procedure itself is fast (in total around 10s for the CIFAR10 samples containing ~700k data points), as the vast majority of the runtime is spent in checking robustness.
> The runtime of this local robustness check heavily depends on the specific oracle chosen.
> The large runtime in table 3 is determined, for instance, by the large computational requirements of the Marabou verifier.
> We have performed additional experiments on MNIST using the LiRPA library used by $\alpha$-$\beta$ CROWN as a local robustness oracle, which significantly improves the total runtime (see reply to `md7J`).
>
> **Different definitions of robustness.** The definition in Kabaha and Cohen [b] essentially corresponds to the definition we use, as global robustness is defined as robustness for those points which are classified with large enough (margin-based) confidence.
> The notion discussed by Wang et. al. [c] is a sensitivity-based notion of global robustness.
> Their definition is not specifically tailored to a classification task, as it does not consider a notion of class explicitly.
> We think our notion is more convenient for a classification task, as our method allows us to provide statements about the robustness conditioned on a specific confidence value.
> Rather, the notion in Wang et. al. [c] captures a notion of function smoothness over all the input space.
> Not too dissimilarly, the definition in Calzavara et. al. [d] considers global robustness as stability in the output on a subset of the input features, together with a label-based notion of robustness.
> We think these definitions are equally viable for their respective tasks, with our specific choice simply being a natural definition for the (classification) problem at hand.
> Our definition only imposes constraints over the behavior of the model for changes that affect the output class, and it is thus a more interesting notion, compared to function smoothness, for classification tasks.
> This class-based setting also gives us access to prediction confidence as a natural indicator of robustness, as both softmax and margin-based confidence are proportional to the required changes in the output-space to change the class.
> We thank you again for the additional references that we will discuss and compare in more detail in a revised version of the manuscript.
>
> Please let us know if we can provide any further details.

---

### Official Review · Reviewer_md7J · 2025-03-17

**Overall Recommendation:** 4

**Summary:**

This method tackles the problem of constructing a method for estimating global robustness of a NN (or other function) that has a formal guarantee. The authors take the approach of probabilistically relaxing the definition of robustness and producing a probabilistic certificate of global robustness. The size of the sample required for the certificate is independent of the input dimension, number of classification classes, etc., which is particularly important for scaling to e.g. deep NNs.

**Claims And Evidence:**

Claims of robustness of certificate are supported by theoretical analysis, see Theorem 4.4 / 4.6.

**Essential References Not Discussed:**

N/A

**Experimental Designs Or Analyses:**

Experiments are sound but could be more impressive and demonstrate scaling, even with a single consumer-grade GPU. MNIST and CIFAR-10 are quite small for 2025 (although no doubt useful for the computationally intensive experiments using formal evaluation).

**Methods And Evaluation Criteria:**

Experiments make sense for the problem at hand. There are ablation studies showing a more confident predictions tend to be more robust under the method proposed in the paper, and that there is a strong dependence between prediction confidence and robustness.

**Other Comments Or Suggestions:**

Typo in Line 162 right column: "For this, we devise on a" => remove "on"

**Other Strengths And Weaknesses:**

N/A

**Questions For Authors:**

Could you please provide additional results showing how the method really scales to larget input dimension and model size?

**Relation To Broader Scientific Literature:**

This work is particularly connected to prior work that defines NN robustness as probability, rather than that relating it to formal verification or distance to adversarial examples. Prior work has shown methods that are can estimate probabilistic robustness either at a given point or globally, but have not had a formal guarantee of robustness.

**Theoretical Claims:**

Yes, checked up to Theorem 4.6.

---

> ### Author Rebuttal · Authors · 2025-03-31
>
> Thank you for your comments. As you suggested we added additional experiments to demonstrate the potential of our approach.
> In our reply, we also address the questions about our experimental evaluation from other reviewers.
>
>
>
> We first point out that
> **the goal of our experiments is to show that our robustness guarantees indeed generalize to unseen data**. Our approach’s complexity scales linearly with the complexity of local robustness checks. We investigated the additional oracles and models proposed by the reviewers.
>
>
> ## New experiments with larger architectures
>
> We focus on MNIST and CIFAR-10, as these are commonly used in related verification literature (as in [1, 2] referred by `8m6m`).
>
> To address scaling, we have performed additional preliminary experiments with bigger architectures for adversarial as well as formal verification.
>
> We run additional experiments with:
> * a VGG network (vgg11_bn [e]), a convolutional architecture with ~10 million parameters, on CIFAR10 with a PGD oracle;
> * ConvBig, a convolutional network used in the references provided by reviewer `8m6m`, on MNIST with a LiRPA (from αβ-CROWN) oracle;
> * the feed forward (FF) network on MNIST (as previously used in the manuscript), with a LiRPA (from αβ-CROWN) oracle.
>
> | Architecture | Training | Oracle | $\epsilon$ | $p_{\min}$| $\delta$ | $\lvert N\rvert$ | $\lvert M\rvert$ | $\lvert\mathbf x_{\kappa\leq\kappa_{\max}}\rvert$ | $n_c$ | $\hat p_\kappa$ | verification runtime (s) | accuracy |
> |--|--|--|--|--|--|--|--|--|--|--|--|--|
> | vgg11_bn | Standard | PGD | $10^{-4}$ |$0.01$| $0.01$ | $685044$ | 10 | 9559 | 0 | 0 | 731 | 0.9174 |
> | ConvBig | DIFF_AI | LiRPA | $2.5\cdot 10^{-3}$|$0.05$ | $0.01$ | $21892$ |12 | 9667 | 4 | 0.000309 | 1838 | 0.9320 |
> | MNIST-FF | AT | LiRPA | $2.5\cdot 10^{-3}$ |$0.05$ |$0.01$ | $21892$  | 7 | 9726 | 1 | 0.000851 | 508 | 0.9328 |
>
> We report architecture, training procedure, the robustness oracle, the size of the mapping $\lvert M \rvert$, the number of predictions for which $\kappa$ is smaller equal than $\kappa_{\max}$ denoted by $\lvert\mathbf x_{\kappa\leq\kappa_{\max}}\rvert$, the estimators $n_c$ and $\hat p_\kappa$ from Sec 6, the runtime of the verification, and the accuracy of the classifier.
> In all our additional experiments, our approach obtains global robustness guarantees that generalize to unseen data, as in all cases $\hat p_{\kappa}<\epsilon/p_{\min}$ and $n_c \leq \lvert D_{\text{test}}\rvert\lvert M\rvert \epsilon$.
> These results demonstrate that our approach works with a variety of datasets, models, and training procedures.
> We will include these results in the revised manuscript.
>
> ## Scalability
>
> The scalability of our method is closely tied to the scalability of the chosen local robustness oracle.
> If results for a single local robustness check can be computed in a time $t$, then we can expect a total runtime of $t\cdot s$, with $s$ the sample complexity obtained from Equation 7.
> PGD scales well in practice: the verification runtime for VGG (~12mins) does not significantly increase in comparison to our results on the different Resnet20 networks (5-20mins, depending on the average robustness of the specific network), even though vgg11_bn has about 40 times as many parameters.
> For some of our experiments with the bigger models, we cannot yet provide results for formal verification with LiRPA, as the GPU hardware requirements of the tool surpass the capabilities of our consumer GPU.
> With reference to the discussion with reviewer `gttu`, we also want to point out that while CROWN verifiers can verify properties on larger networks, the per-sample verification time as reported, for instance, in Wang et al., is still prohibitive to provide global robustness guarantees.
>
> Our experiments show that we obtain good global guarantees that generalize to unseen data for relatively smaller models and input dimensions (28x28x1 grayscale images for MNIST on a smaller network) as well as for larger models and input dimensions (32x32x3 color images on a larger Resnet or VGG network).
>
> References:
> [e] https://github.com/chenyaofo/pytorch-cifar-models

---

### Decision · Program_Chairs · 2025-05-01

**Decision:**

Accept (poster)

**Comment:**

This paper presents a sampling based approach towards certifying global robustness. In particular, the authors attempt to certify the lack of points (or that such regions have low mass) where the model confidence is high but the model prediction is not robust. The authors propose an approach where a certain number of points are sampled and an empirical estimate is constructed. The authors also present an empirical validation of their method when certain popular algorithms such as TRADES are used as robustness oracles for the sample. Overall this is a nice addition to the list of methods on certifying global robustness.